# Excess Risk of Two-Layer ReLU Neural Networks in Teacher-Student Settings and its Superiority to Kernel Methods

**Shunta Akiyama**
Graduate School of Information Science and Technology, The University of Tokyo, Japan
`shunta_akiyama@mist.i-tokyo.ac.jp`

**Taiji Suzuki**
Graduate School of Information Science and Technology, The University of Tokyo, Japan
Center for Advanced Intelligence Project, RIKEN, Japan
`taiji@mist.i.u-tokyo.ac.jp`

## Abstract

While deep learning has outperformed other methods for various tasks, theoretical frameworks that explain its reason have not been fully established. We investigate the excess risk of two-layer ReLU neural networks in a teacher-student regression model, in which a student network learns an unknown teacher network through its outputs. Especially, we consider the student network that has the same width as the teacher network and is trained in two phases: first by noisy gradient descent and then by the vanilla gradient descent. Our result shows that the student network provably reaches a near-global optimal solution and outperforms any kernel methods estimator (more generally, linear estimators), including neural tangent kernel approach, random feature model, and other kernel methods, in a sense of the minimax optimal rate. The key concept inducing this superiority is the non-convexity of the neural network models. Even though the loss landscape is highly non-convex, the student network adaptively learns the teacher neurons.

## 1 Introduction

Explaining why deep learning empirically outperforms other methods has been one of the most significant issues. In particular, from the theoretical viewpoint, it is important to reveal the mechanism of how deep learning trained by an optimization method such as gradient descent can achieve superior generalization performance. To this end, we focus on the excess risk of two-layer ReLU neural networks in a nonparametric regression problem and compare its rate to that of kernel methods. One of the difficulties in showing generalization abilities of deep learning is the non-convexity of the associated optimization problem Li et al. (2018), which may let the solution stacked in a bad local minimum. To alleviate the non-convexity of neural network optimization, recent studies focus on over-parameterization as the promising approaches. Indeed, it is fully exploited by (i) Neural Tangent Kernel (NTK) (Jacot et al., 2018; Allen-Zhu et al., 2019; Arora et al., 2019; Du et al., 2019; Weinan et al., 2020; Zou et al., 2020) and (ii) mean field analysis (Nitanda & Suzuki, 2017; Chizat & Bach, 2018; Mei et al., 2019; Tzen & Raginsky, 2020; Chizat, 2021; Suzuki & Akiyama, 2021).

In the setting of NTK, a relatively large-scale initialization is considered. Then the gradient descent related to parameters of neural networks can be reduced to the convex optimization in RKHS, which is easier to analyze. However, in this regime, it is hard to explain the superiority of deep learning because the estimation ability of the obtained estimator is reduced to that of the corresponding kernel. From this perspective, recent works focus on the "beyond kernel" type analysis Allen-Zhu & Li (2019); Bai & Lee (2020); Li et al. (2020); Chen et al. (2020); Refinetti et al. (2021); Abbe et al. (2022). Although their analysis shows the superiority of deep learning to kernel methods in each setting, in terms of the sample size $(n)$, all derived bounds are essentially $\Omega(1/\sqrt{n})$. This bound is known to be sub-optimal for regression problems Caponnetto & De Vito (2007).

In the mean field analysis setting, a kind of continuous limit of the neural network is considered, and its convergence to some specific target functions has been analyzed. This regime is more suitable in terms of a "beyond kernel" perspective, but it essentially deals with a continuous limit and hence is difficult to control the discretization error when considering a teacher network with a finite width. Indeed, the optimization complexity has been exploited recently in some research, but it still requires an exponential time complexity in the worst case (Mei et al., 2018b; Hu et al., 2019; Nitanda et al., 2021a). This problem is mainly due to the lack of landscape analysis that requires closer exploitation of the problem structure. For example, we may consider the *teacher student setting* where the true function is represented as a neural network. This allows us to use the landscape analysis in the optimization analysis and give a more precise analysis of the statistical performance. In particular, we can obtain a more precise characterization of the *excess risk* (e.g., Suzuki & Akiyama (2021)).

More recently, some studies have focused on the *feature learning* ability of neural networks (Abbe et al., 2021; 2022; Chizat & Bach, 2020; Ba et al., 2022; Nguyen, 2021). Among them, Abbe et al. (2021) considers estimation of the function with staircase property and multi-dimensional Boolean inputs and shows that neural networks can learn that structure through stochastic gradient descent. Moreover, Abbe et al. (2022) studies a similar setting and shows that in a high-dimensional setting, two-layer neural networks with sufficiently smooth activation can outperform the kernel method. However, obtained bound is still $O(1/\sqrt{n})$ and requires a higher smoothness for activation as the dimensionality of the Boolean inputs increases.

The teacher-student setting is one of the most common settings for theoretical studies, e.g., (Tian, 2017; Safran & Shamir, 2018; Goldt et al., 2019; Zhang et al., 2019; Safran et al., 2021; Tian, 2020; Yehudai & Shamir, 2020; Suzuki & Akiyama, 2021; Zhou et al., 2021; Akiyama & Suzuki, 2021) to name a few. Zhong et al. (2017) studies the case where the teacher and student have the same width, shows that the strong convexity holds around the parameters of the teacher network and proposes a special tensor method for initialization to achieve the global convergence to the global optimal. However, its global convergence is guaranteed only for a special initialization which excludes a pure gradient descent method. Safran & Shamir (2018) empirically shows that gradient descent is likely to converge to non-global optimal local minima, even if we prepare a student that has the same size as the teacher. More recently, Yehudai & Shamir (2020) shows that even in the simplest case where the teacher and student have the width *one*, there exist distributions and activation functions in which gradient descent fails to learn. Safran et al. (2021) shows the strong convexity around the parameters of the teacher network in the case where the teacher and student have the same width for Gaussian inputs. They also study the effect of over-parameterization and show that over-parameterization will change the spurious local minima into the saddle points. However, it should be noted that this does not imply that gradient descent can reach the global optima. Akiyama & Suzuki (2021) shows that the gradient descent with a sparse regularization can achieve the global optimal solution for an over-parameterized student network. Thanks to the sparse regularization, the global optimal solution can exactly recover the teacher network. However, this research requires a highly over-parameterized network. Indeed, it requires an exponentially large number of widths in terms of the dimensionality and the sample size. Moreover, they impose quite strong assumptions such that there is no observation noise and the parameter of each neuron in the teacher network should be orthogonal to each other.

The superiority of deep learning against kernel methods has also been discussed in the nonparametric statistics literature. They show the minimax optimality of deep learning in terms of excess risk. Especially a line of research (Schmidt-Hieber, 2020; Suzuki, 2018; Hayakawa & Suzuki, 2020; Suzuki & Nitanda, 2021; Suzuki & Akiyama, 2021) shows that deep learning achieves faster rates of convergence than linear estimators in several settings. Here, the linear estimators are a general class of estimators that includes kernel ridge regression, k-NN regression, and Nadaraya-Watson estimator. Among them, Suzuki & Akiyama (2021) treats a tractable optimization algorithm in a teacher-student setting, but they require an exponential computational complexity and smooth activation function, which does not include ReLU.

In this paper, we consider a gradient descent with two phases, a noisy gradient descent first and a vanilla gradient descent next. Our analysis shows that through this method, the student network recovers the teacher network in a polynomial order computational complexity (with respet to the sample size) without using an exponentially wide network, even though we do not need the strong assumptions such as the no-existence of noise and orthogonality. Moreover, we evaluate the excess risk of the trained network and show that the trained network can outperform any linear estimators,

including kernel methods, in terms of its dependence on the sample size. More specifically, our contributions can be summarized as follows:

- We show that by two-phase gradient descent, the student network, which has the same width as the teacher network, provably reaches the near-optimal solution. Moreover, we conduct a refined analysis of the excess risk and provide the upper bound for the excess risk of the student network, which is much faster than that obtained by the generalization bound analysis with the Rademacher complexity argument. Throughout this paper, our analysis does not require the highly over-parameterization and any special initialization schemes.

- We provide a comparison of the excess risk between the student network and linear estimators and show that while the linear estimators much suffer from the curse of dimensionality, the student network less suffers from that. Particularly, in high dimensional settings, the convergence rate of the excess risk of any linear estimators becomes close to $O(n^{-1/2})$, which coincides with the classical bound derived by the Rademacher complexity argument.

- The lower bound of the excess risk derived in this paper is valid for any linear estimator. The analysis is considerably general because the class of linear estimators includes kernel ridge regression with any kernel. This generality implies that the derived upper bound cannot be derived by the argument that uses a fixed kernel, including Neural Tangent Kernel.

## 2 PROBLEM SETTINGS

**Notations** For $m \in \mathbb{N}$, let $[m] := \{1, \ldots, m\}$. For $x \in \mathbb{R}^d$, $\|x\|$ denotes its Euclidean norm. We denote the inner product between $x, y \in \mathbb{R}^d$ by $\langle x, y \rangle = \sum_{j=1}^d x_i y_i$. $\mathbb{S}^{d-1}$ denotes the unit sphere in $\mathbb{R}^d$. For a matrix $W$, we denote its operator and Frobenius norm by $\|W\|_2$ and $\|W\|_F$, respectively.

Here, we introduce the problem setting and the model that we consider in this paper. We focus on a regression problem where we observe $n$ training examples $D_n = (x_i, y_i)_{i=1}^n$ generated by the following model for an unknown measurable function $f^\circ : \mathbb{R}^d \to \mathbb{R}$:

$$y_i = f^\circ(x_i) + \epsilon_i,$$

where $(x_i)_{i=1}^n$ is independently identically distributed sequence from $P_X$ that is the uniform distribution over $\Omega = \mathbb{S}^{d-1}$, and $\epsilon_i$ are i.i.d. random variables satisfying $\mathbb{E}[\epsilon_i] = 0$, $\mathbb{E}[\epsilon_i^2] = v^2$, and $|\epsilon_i| \leq U$ a.s.. Our goal is to estimate the *true* function $f^\circ$ through the training data. To this end, we consider the square loss $\ell(y, f(x)) = (y - f(x))^2$ and define the expected risk and the empirical risk as $\mathcal{L}(f) := \mathbb{E}_{X,Y}[\ell(Y, f(X)]$ and $\widehat{\mathcal{L}}(f) := \frac{1}{n}\ell(y_i, f(x_i))$, respectively. In this paper, we measure the performance of an estimator $\widehat{f}$ by the *excess risk* $\mathcal{L}(\widehat{f}) - \inf_{f:\text{measurable}} \mathcal{L}(f)$. Since $\inf \mathcal{L}(f) = \mathcal{L}(f^\circ) = 0$, we can check that the excess risk coincides with $\|\widehat{f} - f^\circ\|_{L_2(P_{\mathcal{X}})}^2$, the $L_2$-distance between $\widehat{f}$ and $f^\circ$. We remark that the excess risk is different from the generalization gap $\mathcal{L}(\widehat{f}) - \widehat{\mathcal{L}}(\widehat{f})$. Indeed, when considering the convergence rate with respect to $n$, the generalization gap typically converges to zero with $O(1/\sqrt{n})$ Wainwright (2019). On the other hand, the excess risk can converge with the rate faster than $O(1/\sqrt{n})$, which is known as *fast learning rate*.

### 2.1 MODEL OF TRUE FUNCTIONS

To evaluate the excess risk, we introduce a function class in which the true function $f^\circ$ is included. In this paper, we focus on the teacher-student setting with two-layer ReLU neural networks, in which the true function (called *teacher*) is given by

$$f_{a^\circ, W^\circ}(x) = \sum_{j=1}^m a_j^\circ \sigma(\langle w_j^\circ, x \rangle),$$

where $\sigma(u) = \max\{u, 0\}$ is the ReLU activation, $m$ is the width of the teacher model satisfying $m \leq d$, and $a_j^\circ \in \mathbb{R}$, $w_j^\circ \in \mathbb{R}^d$ for $j \in [m]$ are its parameters. We impose several conditions for the parameters of the teacher networks. Let $W^\circ = (w_1^\circ \ w_2^\circ \ \cdots \ w_m^\circ) \in \mathbb{R}^{d \times m}$ and $\sigma_1 \geq \sigma_2 \geq \cdots \geq \sigma_m$

be the singular values of $W^\circ$. First, we assume that $a_j^\circ \in \{\pm 1\}$ for any $j \in [m]$. Note that by 1-homogeneity of the ReLU activation[1], this condition does not restrict the generality of the teacher networks. Moreover, we assume that there exists $\sigma_{\min} > 0$ such that $\sigma_m > \sigma_{\min}$. If $\sigma_m = 0$, there exists an example in which $f_{a^\circ, W^\circ}$ has multiple representations. Indeed, Zhou et al. (2021) shows that in the case $a_j^\circ = 1$ for all $j \in [m]$ and $\sum w_j^\circ = 0$, it holds that $f_{a^\circ, W^\circ} = \sum_{j=1}^m \sigma(\langle w_j^\circ, x \rangle) = \sum_{j=1}^m \sigma(\langle -w_j^\circ, x \rangle)$. Hence, throughout this paper, we focus on the estimation problem in which the true function is included in the following class:

$$\mathcal{F}^\circ := \{ f_{a^\circ, W^\circ} \mid a^\circ \in \{\pm 1\}^m, \|W^\circ\|_2 \le 1, \sigma_m > \sigma_{\min} \}. \tag{1}$$

This class represents the two-layer neural networks with the ReLU activation whose width is at most the dimensionality of the inputs. The constraint $\|W^\circ\|_2 \le 1$ is assumed only for the analytical simplicity and can be extended to any positive constants.

## 3 ESTIMATORS

In this section, we introduce the classes of estimators: linear estimators and neural networks (student networks) trained by two-phase gradient descent. The linear estimator is introduced as a generalization of the kernel method. We will show separation between *any* linear estimator and neural networks by giving a suboptimal rate of the excess risk for the linear estimators (Theorem 4.1), which simultaneously gives separation between the kernel methods and the neural network approach. A detailed comparison of the excess risk of these estimators will be conducted in Section 4.

### 3.1 LINEAR ESTIMATORS

Given observation $(x_1, y_1), \ldots, (x_n, y_n)$, an estimator $\widehat{f}$ is called *linear* if it is represented by

$$\widehat{f}(x) = \sum_{i=1}^n y_i \varphi_i(x_1, \ldots, x_n, x),$$

where $(\varphi_i)_{i=1}^n$ is a sequence of measurable and $L_2(P_\mathcal{X})$-integrable functions. The most important example in this study is the kernel ridge regression estimator. We note that the kernel ridge estimator is given by $\widehat{f}(x) = Y^\mathsf{T}(K_X + \lambda I)^{-1}\mathbf{k}(x)$, where $K_X = (\mathbf{k}(x_i, x_j))_{i,j=1}^n \in \mathbb{R}^{n \times n}$, $\mathbf{k}(x) = [\mathbf{k}(x, x_1), \ldots, \mathbf{k}(x, x_n)]^\mathsf{T} \in \mathbb{R}^n$ and $Y = [y_1, \ldots, y_n]^\mathsf{T} \in \mathbb{R}^n$ for a kernel function $\mathbf{k} : \mathbb{R}^d \times \mathbb{R}^d \to \mathbb{R}$, which is linear to the output observation $Y$. Since this form is involved in the definition of linear estimators, the kernel ridge regression with any kernel function can be seen as one of the linear estimators. The choice of $\varphi_i$ is arbitrary, and thus the choice of the kernel function is also arbitrary. Therefore, we may choose the best kernel function before we observe the data. However, as we will show in Theorem 4.1, it suffers from a suboptimal rate. Other examples include the $k$-NN estimator and the Nadaraya-Watson estimator. Thus our analysis gives a suboptimality of not only the kernel method but also these well-known linear estimators. Suzuki (2018); Hayakawa & Suzuki (2020) utilized such an argument to show the superiority of deep learning but did not present any tractable optimization algorithm.

### 3.2 STUDENT NETWORKS TRAINED BY TWO-PHASE GRADIENT DESCENT

We prepare the neural network trained through the observation data (called *student*), defined by

$$f(x; \theta) = \sum_{j=1}^m a_j \sigma(\langle w_j, x \rangle),$$

where $\theta = ((a_1, w_1), \ldots (a_m, w_m)) \in \mathbb{R}^{(d+1)m} =: \Theta$. We assume that the student and teacher networks have the same width. Based on this formulation, we aim to train the parameter $\theta$ that will be provably close to that of the teacher network. To this end, we introduce the training algorithm, two-phase gradient descent, which we consider in this paper.

---

[1] $\sigma(\langle w, x \rangle) = \|w\| \sigma(\langle w/\|w\|, x \rangle)$ for any $w \in \mathbb{R}^d/\{\mathbf{0}\}$ and $x \in \mathbb{R}^d$.

**Phase I: noisy gradient descent** For $r \in \mathbb{R}$, let $\bar{r} := R \cdot \tanh\left(r|r|/2R\right)$ be a clipping of $r$, where $R > 1$ is a fixed constant. In the first phase, we conduct a noisy gradient descent with the weight decay regularization. The objective function used to train the student network is given as follows:

$$\widehat{\mathcal{R}}_\lambda(\theta) := \frac{1}{2n}\sum_{i=1}^{n}(y_i - f(x_i; \bar{\theta}))^2 + \lambda\sum_{j=1}^{m}\left(|a_j|^2 + \|w_j\|^2\right),$$

where $\bar{\theta}$ is the element-wise clipping of $\theta$ and $\lambda > 0$ is a regularization parameter. The parameter clipping ensures the bounded objective value and smoothness of the expected risk around the origin, which will be helpful in our analysis. Then, the parameters of the student network are updated by

$$\theta^{(k+1)} = \theta^{(k)} - \eta^{(1)}\nabla\widehat{\mathcal{R}}_\lambda\left(\theta^{(k)}\right) + \sqrt{\frac{2\eta^{(1)}}{\beta}}\zeta^{(k)},$$

where $\eta^{(1)} > 0$ is a step-size, $\left\{\zeta^{(k)}\right\}_{k=1}^{\infty}$ are independently identically distributed noises from the standard normal distribution, and $\beta > 0$ is a constant called *inverse temperature*. This type of noisy gradient descent is called *gradient Langevin dynamics*. It is known that by letting $\beta$ be large, we can ensure that the smooth objective function will decrease. On the other hand, because of the non-smoothness of the ReLU activation, the objective function $\widehat{\mathcal{R}}_\lambda$ is also non-smooth. Hence it is difficult to guarantee the small objective value. To overcome this problem, we evaluate the expected one instead in the theoretical analysis, which is given by

$$\mathcal{R}_\lambda(\theta) := \frac{1}{2}\mathbb{E}_x\left[\left(f_{a^\circ, W^\circ}(x) - f(x; \bar{\theta})\right)^2\right] + \lambda\sum_{j=1}^{m}\left(|a_j|^2 + \|w_j\|^2\right).$$

We can ensure a small $\mathcal{R}_\lambda(\theta)$ after a sufficient number of iterations (see Section 4.2 for the detail).

**Phase II: vanilla gradient descent** After phase I, we can ensure that for each node of the student network, there is a node of the teacher network that is relatively close to each other. Then we move to conduct the vanilla gradient descent to estimate the parameters of the teacher more precisely. Before conducting the gradient descent, we rescale the parameters as follows:

$$a_j^{(k)} \leftarrow \text{sgn}(\bar{a}^{(k)}), \ w_j^{(k)} \leftarrow \left|\bar{a}^{(k)}\right|\bar{w}_j^{(k)}, \qquad \forall j \in [m].$$

We note this transformation does not change the output of the student network thanks to the 1-homogeneity of the ReLU activation. After that, we update the parameters of the first layer by

$$W^{(k+1)} = W^{(k)} - \eta^{(2)}\nabla_W\widehat{\mathcal{R}}\left(W^{(k)}\right),$$

where $\eta^{(2)} > 0$ is a step-size different from $\eta^{(1)}$ and $\widehat{\mathcal{R}}(W) := \frac{1}{2n}\sum_{i=1}^{n}(y_i - f(x_i; \theta))^2$. In this phase, we no longer need to update the parameters of both layers. Moreover, the regularization term and the gradient noise added in phase I are also unnecessary. These simplifications of the optimization algorithm are based on the strong convexity of $\widehat{\mathcal{R}}(W)$ around $W^\circ$, the parameters of the teacher network. The analysis for this local convergence property is based on that of Zhang et al. (2019), and eventually, we can evaluate the excess risk of the student network.

The overall training algorithm can be seen in Algorithm 1. In summary, we characterize the role of each phase as follows: in phase I, the student network explore the parameter space globally and finds the parameters that are relatively close to that of teachers, and in phase II, the vanilla gradient descent for the first layer outputs more precise parameters, as we analyze in Section 4.2.

**Remark 3.1.** *Akiyama & Suzuki (2021) also considered the convergence of the gradient descent in a teacher-student model. They considered a sparse regularization, $\sum_{j=1}^{m}|a_j|\|w_j\|$, for the ReLU activation while we consider the $L_2$-regularization given by $\sum_{j=1}^{m}(|a_j|^2 + \|w_j\|^2)$. These two regularizations are essentially the same since the minimum of the later regularization under the constraint of $|a_j|\|w_j\| = \text{const.}$ is given by $2\sum_{j=1}^{m}|a_j|\|w_j\|$ by the arithmetic-geometric mean relation. On the other hand, Akiyama & Suzuki (2021) consider a vanilla gradient descent instead of the noisy gradient descent. This makes it difficult to reach the local region around the optimal solution, and their analysis required an exponentially large width to find the region. We may use a narrow network in this paper with the same width as the teacher network. This is due to the ability of the gradient Langevin dynamics to explore the entire space and find the near global optimal solution.*

---

**Algorithm 1** Two-Phase Gradient Descent

---

**Input:** max iteration $k^{(1)}$ and $k^{(2)}$, stepsize parameter $\eta^{(1)}$, $\eta^{(2)} > 0$, regularization parameter $\lambda > 0$, inverse temperature $\beta > 0$.
1: *Initialization*: $\theta^{(0)} \sim \rho_0$.
2: **for** $k = 1, 2, \ldots, k^{(1)}$ **do**
3:    $\zeta^{(k)} \sim \mathcal{N}(0, I_{m(d+1)})$
4:    $\theta^{(k+1)} = \theta^{(k)} - \eta^{(1)}\nabla\widehat{\mathcal{R}}_\lambda\big(\theta^{(k)}\big) + \sqrt{\frac{2\eta^{(1)}}{\beta}}\zeta^{(k)}$
5: **end for**
6: Reparameterization: $a_j^{(k)} = \text{sgn}(\bar{a}_j^{(k)})$, $w_j^{(k)} = \left|\bar{a}_j^{(k)}\right|\bar{w}_j^{(k)}$
7: **for** $k = k^{(1)} + 1, k^{(1)} + 2, \ldots, k^{(2)}$ **do**
8:    $W^{(k+1)} = W^{(k)} - \eta^{(2)}\nabla_W\widehat{\mathcal{R}}\big(W^{(k)}\big)$
9: **end for**

---

## 4   EXCESS RISK ANALYSIS AND ITS COMPARISON

This section provides the excess risk bounds for linear estimators and the deep learning estimator (the trained student network). More precisely, we give its lower bound for linear estimators and upper bound for the student network. As a consequence, it will be provided that the student network achieves a faster learning rate and less hurt from a curse of dimensionality than linear estimators.

### 4.1   MINIMAX LOWER BOUND FOR LINEAR ESTIMATORS

Here, we analyze the excess risk of linear estimators and introduce its lower bound. More specifically, we consider the minimax excess risk over the class of linear estimators given as follows:

$$R_{\text{lin}}(\mathcal{F}^\circ) = \inf_{\widehat{f}:\text{linear}} \sup_{f^\circ \in \mathcal{F}^\circ} \mathbb{E}_{D_n}[\|\widehat{f} - f^\circ\|_{L_2(P_\mathcal{X})}^2],$$

where the infimum is taken over all linear estimators, and the expectation is taken for the training data. This expresses the infimum of worst-case error over the class of linear estimators to estimate a function class $\mathcal{F}^\circ$. Namely, any class of linear estimators cannot achieve a faster excess risk than $R_{\text{lin}}(\mathcal{F}^\circ)$. Based on this concept, we provide our result about the excess risk bound for linear estimators. Under the definition of $\mathcal{F}^\circ$ by Eq. (1), we can obtain the lower bound as follows:

**Theorem 4.1.** *For arbitrary small $\kappa > 0$, we have that*

$$R_{\text{lin}}(\mathcal{F}^\circ) \gtrsim n^{-\frac{d+2}{2d+2}}n^{-\kappa}.$$

The proof can be seen in Appendix A. This theorem implies that under $d \geq 2$, the convergence rate of excess risk is at least slower than $n^{-\frac{2+2}{2\cdot 2+2}} = n^{-2/3}$. Moreover, since $-\frac{d+2}{2d+2} \to -1/2$ as $d \to \infty$, the convergence rate of excess risk will be close to $n^{-1/2}$ in high dimensional settings, which coincides with the generalization bounds derived by the Rademacher complexity argument. Hence, we can conclude that the linear estimators suffer from the curse of dimensionality.

The key strategy to show this theorem is the following "convex-hull argument" given as follows:

$$R_{\text{lin}}(\mathcal{F}^\circ) = R_{\text{lin}}(\overline{\text{conv}}(\mathcal{F}^\circ)),$$

where $\text{conv}(\mathcal{F}^\circ) := \{\sum_{j=1}^N \lambda_j f_j \mid N \in \mathbb{N}, f_j \in \mathcal{F}^\circ, \lambda_j \geq 0, \sum_{j=1}^N \lambda_j = 1\}$ and $\overline{\text{conv}}(\cdot)$ is the closure of $\text{conv}(\cdot)$ in $L_2(P_\mathcal{X})$. By combining this argument with the minimax optimal rate analysis exploited in Zhang et al. (2002) for linear estimators, we obtain the rate in Theorem 4.1.

This equality implies that the linear estimators cannot distinguish the original class $\mathcal{F}^\circ$ and its convex hull $\overline{\text{conv}}(\mathcal{F}^\circ)$. Therefore, if the function class $\mathcal{F}^\circ$ is highly non-convex, then the linear estimators result in a much slower convergence rate since $\overline{\text{conv}}(\mathcal{F}^\circ)$ will be much larger than that of the original class $\mathcal{F}^\circ$. Indeed, we can show that the convex hull of the teacher network class is *considerably larger* than the original function class, which causes the curse of dimensionality. For example, the mean of two teacher networks with a width $m$ can be a network with width $2m$, which shows that $\text{conv}(\mathcal{F}^\circ)$ can consist of much wider networks. See Appendix A for more details.

## 4.2 EXCESS RISK OF THE NEURAL NETWORKS

Here, we give an upper bound of the excess risk of the student network trained by Algorithm 1. The main result is shown in Theorem 4.6, which states that the student network can achieve the excess risk with $O(n^{-1})$. This consequence is obtained by three-step analysis. First, (1) we provide a convergence guarantee for phase I and phase II in Algorithm 1. We first show that by phase I, the value of $\mathcal{R}_\lambda(\theta^{(k)})$ will be sufficiently small (see Proposition 4.3). Then, (2) we can show that the parameters of the student network and the teacher networks are close to each other by Proposition 4.4. By using the strong convexity around the parameters of the teacher network, the convergence of phase II is ensured. By combining these result, (3) we get the excess risk bound as Theorem 4.6.

**(1) Convergence in phase I:** First, we provide a convergence result and theoretical strategy of the proof for phase I. Since the ReLU activation is non-smooth, the loss function $\widehat{\mathcal{R}}_\lambda(\cdot)$ is also non-smooth. Therefore it is difficult to ensure the convergence of the gradient Langevin dynamics. To overcome this problem, we evaluate the value of $\mathcal{R}_\lambda(\cdot)$ instead by considering the update $\theta^{(k+1)} = \theta^{(k)} - \eta^{(1)}\nabla\mathcal{R}_\lambda(\theta^{(k)}) + \sqrt{\frac{2\eta^{(1)}}{\beta}}\zeta^{(k)}$, and bound the residual due to using the gradient of $\widehat{\mathcal{R}}_\lambda(\cdot)$. This update can be interpreted as the discretization of the following stochastic differential equation:

$$\mathrm{d}\theta = -\beta\nabla\mathcal{R}_\lambda(\theta)\mathrm{d}t + \sqrt{2}\mathrm{d}B_t,$$

where $(B_t)_{t\geq 0}$ is the standard Brownian motion in $\Theta(= \mathbb{R}^{(d+1)m})$. It is known that this process has a unique invariant distribution $\pi_\infty$ that satisfies $\frac{\mathrm{d}\pi_\infty}{\mathrm{d}\theta}(\theta) \propto \exp(-\beta\mathcal{R}_\lambda(\theta))$. Intuitively, as $\beta \to \infty$, this invariant measure concentrates around the minimizer of $\mathcal{R}_\lambda$. Hence, by letting $\beta$ sufficiently large, obtaining a near-optimal solution will be guaranteed.

Such a technique for optimization is guaranteed in recent works (Raginsky et al., 2017; Erdogdu et al., 2018). However, as we stated above, they require a smooth objective function. Therefore we cannot use the same technique here directly. To overcome this difficulty, we evaluate the difference between $\nabla\widehat{\mathcal{R}}_\lambda$ and $\nabla\mathcal{R}_\lambda$ as follows:

**Lemma 4.2.** *There exists a constant $C > 0$ such that with probability at least $1 - \delta$, it holds that*

$$V_{grad} := \sup_\theta \left\| \nabla\mathcal{R}_\lambda(\theta) - \nabla\widehat{\mathcal{R}}_\lambda(\theta) \right\| \leq CR^3 m \sqrt{\frac{d\log(mdn/\delta)}{n}}.$$

This lemma implies that with high probability, the difference between $\nabla\widehat{\mathcal{R}}_\lambda$ and $\nabla\mathcal{R}_\lambda$ will vanish as $n \to \infty$. Thanks to this lemma, we can connect the dynamics of the non-smooth objective with that of the smooth objective and import the convergence analysis developed so far in the smooth objective. In particular, we utilize the technique developed by Vempala & Wibisono (2019) (see Appendix C for more details). We should note that our result extends the existing one Vempala & Wibisono (2019) in the sense that it gives the convergence for the non-differential objective function $\widehat{\mathcal{R}}_\lambda(\cdot)$. As a consequence, we obtain the following convergence result as for phase I.

**Proposition 4.3.** *Let $\mathcal{R}_\lambda^*$ be the minimum value of $\mathcal{R}_\lambda$ in $\Theta$, $q$ be a density function of $\pi_\infty$ (i.e., $q(\theta) \propto \exp(-\beta\mathcal{R}_\lambda(\theta))$) and $H_q(p) := \int_\mathbb{R} p(\theta)\log\frac{p(\theta)}{q(\theta)}\mathrm{d}x$ be the KL-divergence. There exists a constant $c, C > 0$ and the log-Sobolev constant $\alpha$ (defined in Lemma C.4) such that with step-size $0 < \eta^{(1)} < c\frac{\delta\lambda\alpha}{\beta R^3 m^3 d}$, after $k^{(1)} \geq \frac{\beta}{\alpha\eta^{(1)}}\log\frac{2H_q(\rho_0)}{\delta}$ iteration, the output $\theta^{(k)}$ satisfies*

$$\mathbb{E}[\mathcal{R}_\lambda(\theta^{(k)})] - \mathcal{R}_\lambda^* \leq C\left[(\lambda + m)\exp(m^2\beta)\sqrt{\delta + \frac{1}{3n\lambda}} + \frac{d}{2\beta}\log\left(\frac{m^3 d\beta}{\lambda}\right)\right]$$

*with probability at least $1 - \delta$, where the expectation is taken over the initialization and Gaussian random variables added in the algorithm.*

Therefore, we can see that phase I optimization can find a *near optimal solution* with a polynomial time complexity (with respect to $n$) even though the objective function is non-smooth. It also may be considered to use the gradient Langevin dynamics to reach the global optimal solution by using higher $\beta$. However, it requires increasing the inverse temperature $\beta$ exponentially related to $n$ and other parameters, which leads to exponential computational complexity. To overcome this difficulty,

we utilize the local landscape of the objective function. We can show the objective function will be strongly convex around the teacher parameters and we do not need to use the gradient noise and any regularization. Indeed, we can show that the vanilla gradient descent can reach the global optimal solution in phase II, as shown in the following.

**(2) Convergence in phase II:**   Next, we prove the convergence guarantee of phase II and provide an upper bound of the excess risk. The convergence result is based on the fact that when $\mathcal{R}_\lambda(\theta)$ is small enough (guaranteed in Proposition 4.3), the parameters of the student network will be close to those of the teacher network, as the following proposition:

**Proposition 4.4.** *There exists a threshold $\epsilon_0 = \mathrm{poly}(m^{-1}, \sigma_{\min})$ such that by letting $\lambda \le \epsilon_0/m$, if $\epsilon = \mathcal{R}_\lambda(\theta) - \mathcal{R}_\lambda^* \le \epsilon_0$, it holds that for every $j \in [m]$, there exists $k_j \in [m]$ such that $\mathrm{sgn}(a_{k_j}) = a_j^\circ$ and $\left\| |a_{k_j}| w_{k_j} - w_j^\circ \right\| \le c\sigma_m/\kappa^3 m^3$.*

The proof of this proposition can be seen in Appendix D. We utilize the technique in Zhou et al. (2021), which give the same results to the cases when the activation is the absolute value function. In this proposition, we compare the parameters of the teacher network with the *normalized* student parameters. This normalization is needed because of the 1-homogeneity of the ReLU activation. The inequality $\left\| |a_{k_j}| w_{k_j} - w_j^\circ \right\| \le c\sigma_m/\kappa^3 m^3$ ensures the closeness of parameters in the sense of the *direction* and the *amplitude*. Combining this with the equality $\mathrm{sgn}(a_{k_j}) = a_j^\circ$, we can conclude the closeness and move to ensure the local convergence. Thanks to this closeness and local strong convexity, we can ensure the convergence in phase II as follows:

**Lemma 4.5.** *Let $\kappa := \sigma_1/\sigma_m$ and $\tilde{\sigma} := (\prod_{j=1}^m \sigma_j)/\sigma_m^m$. Suppose that the condition is Proposition 4.4 holds. Then there exists absolute constants $c_1$, $c_2$, $c_3$, $c_4$, $c_5$ such that under*

$$n \ge \frac{c_1 \kappa^{10} m^9 d}{\sigma_m} \log\left(\frac{\kappa m d}{\sigma_m}\right) \cdot \left(\|W^*\|_F^2 + v^2\right),$$

*the output of the gradient descent with step-size $\eta \le \frac{1}{c_2 \kappa m^2}$ satisfies*

$$\|\widehat{f} - f^\circ\|_{L_2(P_\mathcal{X})}^2 \lesssim \frac{\tilde{\sigma}^2 \sigma_{\min}^{-4} m^5 \log n}{n} + c_4 \left(1 - \frac{c_3 \eta}{\tilde{\sigma}\kappa^2}\right)^k \cdot \frac{\sigma_m}{\kappa^3 m^2}$$

*after $k$ iterations with probability at least $1 - c_5 d^{-10}$.*

**(3) Unified risk bound:**   By combining (1) and (2), we obtain a unified result as follows:

**Theorem 4.6.** *There exists $\epsilon_0 = \mathrm{poly}(m^{-1}, \sigma_{\min})$ and constants $C$ and $C' > 0$ such that for any $0 < b < 1$, under $n \ge \lambda \sigma_b^{-3} \exp(\sigma_b^{-1} m^2)$ where $\sigma_b := b\epsilon_0$, let $k^{(1)} = C\lambda^{-2}\beta^{-1}\exp(m^2\beta)$ and $k^{(2)} = k^{(1)} + \log(C' n \eta^{(2)-2})$, then the output of Algorithm 1 with $\lambda = \sigma_b d^{-1}$, $\beta = \Omega(\sigma_b^{-1} d)$, $\eta^{(1)} = O(\lambda \sigma_b^3 \exp(\sigma_b^{-1} m^2))$ and $\eta^{(2)} = O(\sigma_{\min} m^{-2})$ satisfies*

$$\|\widehat{f} - f^\circ\|_{L_2(P_\mathcal{X})}^2 \lesssim \frac{\tilde{\sigma}^2 \sigma_{\min}^{-4} m^5 \log n}{n}$$

*with probability at least $1 - b - d^{-10}$, where $\tilde{\sigma} = (\prod_{j=1}^m \sigma_j)/\sigma_m^m$.*

The proof of this theorem also can be seen in Appendix D. This theorem implies that for fixed $m$, the excess risk of the student networks is bounded by

$$\mathbb{E}_{D_n}[\|\widehat{f} - f^\circ\|_{L_2(P_\mathcal{X})}^2] \lesssim n^{-1}.$$

As compared to the lower bound derived for linear estimators in Theorem 4.1, we get the faster rate $n^{-1}$ in terms of the sample size. Moreover, the dependence of the excess risk on the dimensionality $d$ does not appear explicitly. Therefore we can conclude that the student network less suffers from the curse of dimensionality than linear estimators. As we pointed out in the previous subsection, the convex hull argument causes the curse of dimensionality for linear estimators since they only prepare a fixed basis. On the other hand, the student network can "find" the basis of the teacher network via noisy gradient descent in phase I and eventually avoid the curse of dimensionality.

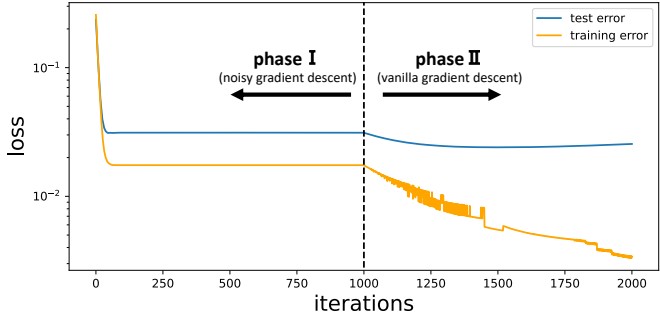

Figure 1: Convergence of the training loss and test loss.

**Remark 4.7.** *Akiyama & Suzuki (2021) establishes the local convergence theory for the student wider than the teacher. However, it cannot apply here since they only consider the teacher whose parameters are orthogonal to each other. Suzuki & Akiyama (2021) also showed the benefit of the neural network and showed the superiority of deep learning in a teacher-student setting where the teacher has infinite width. They assume that the teacher has decaying importance; that is, the teacher can be written as $f^\circ(x) = \sum_{j=1}^{\infty} a_j^\circ \sigma(\langle w_j^\circ, x \rangle)$ where $a_j^\circ \lesssim j^{-a}$ and $w_j^\circ \lesssim j^{-b}$ (with an exponent $a, b > 0$) for a bounded smooth activation $\sigma$. Our analysis does not assume the decay of importance, and the activation function is the non-differential ReLU function. Moreover, Suzuki & Akiyama (2021) considers a pure gradient Langevin dynamics instead of the two-stage algorithm, which results in the exponential computational complexity in contrast to our analysis.*

## 5 NUMERICAL EXPERIMENT

In this section, we conduct a numerical experiment to justify our theoretical results. We apply Algorithm 1 to the settings $d = m = 50$. For the teacher network, we employ $a_j^\circ = 1$ for $1 \le j \le 25$, $a_j^\circ = -1$ for $26 \le j \le 50$ and $(w_1^\circ, \ldots, w_{50}^\circ) = I_{50}$ as its parameters. The parameters of the student network are initialized by $\theta^{(0)} \sim \mathcal{N}(0, I_{m(d+1)})$. We use the sample with $n = 1000$ as the training data. Hyperparameters are set to $\eta^{(1)} = \eta^{(2)} = 0.01$, $\beta = 100$, $\lambda = 0.01$, $k_{\max}^{(1)} = 1000$ and $k_{\max}^{(2)} = 2000$. Figure 1 shows the experimental result. The orange line represents the training loss without the regularization term. The blue line represents the test loss. Since we can compute the generalization error analytically (see Appendix B), we utilize its value as the test loss.

We can see that in phase I, both the training and test losses decrease first and then fall flat. At the beginning of phase II, we can observe that both the training and test losses decrease linearly. This reflects the strong convexity around the parameters of the teacher network, as we stated in the convergence guarantee of phase II. While the training loss keeps going up and down, the curve of the test loss is relatively smooth. This difference is due to the smoothness of the generalization loss (or $\mathcal{R}_\lambda$), which we use in the convergence analysis in phase I. The test loss does not keep decreasing and converges to a constant. The existence of the sample noise causes this phenomenon: even if the parameters of the student coincide with that of the teacher, its training loss will not be zero. Thus we can say that the numerical experiment is consistent with our theoretical results.

## 6 CONCLUSION

In this paper, we focus on the nonparametric regression problem, in which a true function is given by a two-layer neural network with the ReLU activation, and evaluate the excess risks of linear estimators and neural networks trained by two-phase gradient descent. Our analysis revealed that while any linear estimator suffers from the curse of dimensionality, deep learning can avoid it and outperform linear estimators, which include the neural tangent kernel approach, random feature model, and other kernel methods. Essentially, the non-convexity of the model induces this difference.

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

## A PROOF OF THEOREM 4.1

First, we introduce the formal statement of the "convex hull argmunent" we stated in Section 4.1.

**Proposition A.1** (Hayakawa & Suzuki (2020)). *The minimax optimal rate of linear estimators on a target function class $\mathcal{F}^\circ$ is the same as that on the convex hull of $\mathcal{F}^\circ$:*

$$R_{\mathrm{lin}}(\mathcal{F}^\circ) = R_{\mathrm{lin}}(\overline{\mathrm{conv}}(\mathcal{F}^\circ)),$$

*where* $\mathrm{conv}(\mathcal{F}^\circ) \coloneqq \{\sum_{j=1}^N \lambda_j f_j \mid N \in \mathbb{N}, f_j \in \mathcal{F}^\circ, \lambda_j \geq 0, \sum_{j=1}^N \lambda_j = 1\}$ *and* $\overline{\mathrm{conv}}(\cdot)$ *is the closure of* $\mathrm{conv}(\cdot)$ *in* $L_2(P_\mathcal{X})$.

For the proof, we use this convex hull argument and the minimax optimal rate analysis for linear estimators developed by Zhang et al. (2002). They essentially showed the following statement in their Theorem 1. Note that they consider the class of linear estimators on the Euclidean space, but we can apply the same argument for the class of linear estimators on $\mathbb{S}^{d-1}$.

**Proposition A.2** (Theorem 1 of Zhang et al. (2002)). *Let $\mu$ be uniform measure on $\mathbb{S}^{d-1}$ satisfying $\mu(\mathbb{S}^{d-1}) = 1$. Suppose that the space $\Omega$ has even partition $\mathcal{A}$ such that $|\mathcal{A}| = 2^K$ for an integer $K \in \mathbb{N}$, each $A \in \mathcal{A}$ has measure $\alpha_1 2^{-K} \leq \mu(A) \leq \alpha_2 2^{-K}$ for constants $\alpha_1, \alpha_2 > 0$, and $\mathcal{A}$ is indeed a partition of $\Omega$, i.e., $\cup_{A \in \mathcal{A}} A = \Omega$, $A \cap A' = \emptyset$ for $A, A' \in \mathcal{A}$ and $A \neq A'$. Then, if $K$ is chosen as $n^{-\gamma_1} \leq 2^{-K} \leq n^{-\gamma_2}$ for constants $\gamma_1, \gamma_2 > 0$ that are independent of $n$, then there exists an event $\mathcal{E}$ such that, for a constant $C' > 0$,*

$$P(\mathcal{E}) \geq 1 - o(1) \text{ and } |\{x_i \mid x_i \in A \ (i \in \{1, \dots, n\})\}| \leq C' \alpha_2 n 2^{-K} \ (\forall A \in \mathcal{A}).$$

*Moreover, suppose that, for a class $\mathcal{F}^\circ$ of functions on $\Omega$, there exists $\Delta > 0$ that satisfies the following conditions:*

1. *There exists $F > 0$ such that, for any $A \in \mathcal{A}$, there exists $g \in \mathcal{F}^\circ$ that satisfies $g(x) \geq \frac{1}{2}\Delta F$ for all $x \in A$,*

2. *There exists $K'$ and $C'' > 0$ such that $\frac{1}{n}\sum_{i=1}^n g(x_i)^2 \leq C'' \Delta^2 2^{-K'}$ for any $g \in \mathcal{F}^\circ$ on the event $\mathcal{E}$.*

*Then, there exists a constant $F_1$ such that at least one of the following inequalities holds:*

$$\frac{F^2}{4F_1 C''} \frac{2^{K'}}{n} \leq R_{\mathrm{lin}}(\mathcal{F}^\circ),$$

$$\frac{F^3}{32}\Delta^2 2^{-K} \leq R_{\mathrm{lin}}(\mathcal{F}^\circ),$$

*for sufficiently large $n$.*

**Lemma A.3.** *Let $0 < \Delta \leq 1/2$ and let $g : \mathbb{S}^{d-1} \to \mathbb{R}$ be a function defined by*

$$g(x) = \frac{1}{d-1}\sum_{j=2}^d \left[ -\sigma(x_j) + \frac{1}{2}\sigma(x_j + 2\Delta \cdot x_1) + \frac{1}{2}\sigma(x_j - 2\Delta \cdot x_1) \right].$$

*Then it holds that $g(x) \geq \Delta/2$ for $x \in \mathbf{B}_\Delta^\infty(\mathbf{e}_1)$ and $g(x) = 0$ for $x \notin \mathbf{B}_{2\Delta}^\infty(\mathbf{e}_1)$, where $\mathbf{e}_1 \coloneqq (1, 0, \dots, 0) \in \mathbb{S}^{d-1}$ and $\mathbf{B}_r^\infty(\mathbf{e}_1) \coloneqq \{x \in \mathbb{S}^{d-1} \mid \|x - \mathbf{e}_1\|_\infty \leq r\}$ for $r > 0$.*

*Proof.* Let $g_j(x) = -\sigma(x_j) + \frac{1}{2}\sigma(x_j + 2\Delta \cdot x_1) + \frac{1}{2}\sigma(x_j - 2\Delta \cdot x_1)$. First we suppose that $x \in \mathbf{B}_\Delta^\infty$. Then, we have $x_1 \geq 1 - \Delta$ and $|x_j| \leq \Delta$ for any $j \in \{2, \dots, d\}$. If $0 \leq x_j \leq \Delta$, it holds that

$$g_j(x) = -\sigma(x_j) + \frac{1}{2}\sigma(x_j + 2\Delta \cdot x_1) + \frac{1}{2}\sigma(x_j - 2\Delta \cdot x_1) = \frac{1}{2}(2\Delta \cdot x_1 - x_j) \geq \Delta(1 - \Delta) \geq \frac{1}{2}\Delta.$$

Moreover, if $-\Delta \leq x_j \leq 0$, we get

$$g_j(x) = -\sigma(x_j) + \frac{1}{2}\sigma(x_j + 2\Delta \cdot x_1) + \frac{1}{2}\sigma(x_j - 2\Delta \cdot x_1) = \frac{1}{2}(x_j + 2\Delta \cdot x_1) \geq \Delta(1 - \Delta) \geq \frac{1}{2}\Delta.$$

Hence, we get the first assertion by $g(x) = \frac{1}{d-1}\sum_{j=2}^d g_j(x) \geq \frac{\Delta}{2}$.

Next we suppose $x \notin \mathbf{B}_{2\Delta}^{\infty}(\mathbf{e}_1)$. Then, it holds that $x_j \geq 2\Delta \geq 2\Delta x_1$ for any $j \in \{2, \ldots, d\}$. Hence it holds that $|x_j/x_1| \geq 2\Delta$, and we obtain that $\mathrm{sgn}(x_j + 2\Delta \cdot x_1) = \mathrm{sgn}(x_j) = \mathrm{sgn}(x_j - 2\Delta \cdot x_1) \in \{\pm 1\}$. We can check $g_j(x) = 0$ for each case, and hence it holds that $g(x) = 0$. Thus we get the second assertion. $\square$

*proof of Theorem 4.1.* Let us consider the covering of $\mathbb{S}^{d-1}$ by spherical caps, i.e., $\mathbf{B}_r(x) \cap \mathbb{S}^{d-1}$ for some $x \in \mathbb{S}^{d-1}$ with radius $r$ satisfying $r \in (0, 1)$. It is known that there is a covering $\mathcal{A}$ with $|\mathcal{A}| \sim r^{-d}$ (ignoring logarithm terms). Then, by letting $r \sim 2^{-K/d}$, there exists a covering $\mathcal{A}$ satisfying $|\mathcal{A}| = 2^K$.

For each $A \in \mathcal{A}$, we define a function $g_A$ by the same manner as in Lemma A.3, i.e., for $A \in \mathcal{A}$ written by $\mathbf{B}_r(x_A) \cap \mathbb{S}^{d-1}$ with $x_A \in \mathbb{S}^{d-1}$, we consider the orthogonal basis including $x_A$ and define $g_A$ with regrading $x_A$ as $\mathbf{e}_1$. Define $\mathcal{F}_{\mathcal{A}}^{\circ} := \{g_A/2 \mid A \in \mathcal{A}\}$. It is not difficult to check that $\mathcal{F}_{\mathcal{A}}^{\circ} \in \overline{\mathrm{conv}}(\mathcal{F}^{\circ})$. Then by Proposition A.1, it holds that

$$R_{\mathrm{lin}}(\mathcal{F}^{\circ}) = R_{\mathrm{lin}}(\overline{\mathrm{conv}}(\mathcal{F}^{\circ})) \geq R_{\mathrm{lin}}(\mathcal{F}_{\mathcal{A}}^{\circ}),$$

where the inequality follows from $\mathcal{F}_{\mathcal{A}}^{\circ} \in \overline{\mathrm{conv}}(\mathcal{F}^{\circ})$. Hence, it suffices to give the lower bound for the right hand side. Now, we apply Proposition A.2 with $\mathcal{F}^{\circ} = \mathcal{F}_{\mathcal{A}}^{\circ}$ and $K = K'$. Applying Lemma A.3 with $\Delta = 2^{-K/d}$ In the event $\mathcal{E}$ which we introduce in Proposition A.2, there exists a constant $C'$ such that $|\{x_i \mid x_i \in A \ (i \in \{1, \ldots, n\})\}| \leq C'\alpha_2 n 2^{-K}$ for all $A \in \mathcal{A}$. Therefore, we obtain that

$$\frac{1}{n}\sum_{i=1}^{n} g_A(x_i)^2 \lesssim \frac{1}{n}n 2^{-K} \cdot \Delta^2 = 2^{-K}\Delta^2$$

Therefore, Proposition A.2 gives $R_{\mathrm{lin}}(\mathcal{F}_{\mathcal{A}}^{\circ}) \gtrsim \min\{\frac{2^K}{n}, 2^{-(1+2/d)K}\}$. By letting $2^K \sim n^{d/2(d+1)}$, we get the assertion. $\square$

## B   EXPLICIT FORM OF THE OBJECTIVE FUNCTION AND ITS GRADIENT

In this section, we derive the explicit form of $\mathcal{R}_\lambda(\cdot)$ and its gradient, which we utilize in our analysis (especially that of the convergence in phase I). First, for $w, v \in \mathbb{R}^d/\{\mathbf{0}\}$, we have that

$$\mathbb{E}_{x \sim P_X}[\sigma(\langle w, x \rangle)\sigma(\langle v, x \rangle)] = \frac{\mathbb{E}_{\tilde{x} \sim \mathcal{N}(0, I_d)}[\sigma(\langle w, x \rangle)\sigma(\langle v, x \rangle)]}{\mathbb{E}_{\tilde{x} \sim \mathcal{N}(0, I_d)}[\|\tilde{x}\|^2]}$$

$$= \frac{\sin\phi(w, v) + (\pi - \phi(w, v))\cos\phi(w, v)}{2\pi d}\|w\|\|v\|, \qquad (2)$$

where $\phi(w, v) := \arccos(\langle w, v \rangle/\|w\|\|v\|)$. The second equality follows from $\mathbb{E}_{\tilde{x} \sim \mathcal{N}(0, I_d)}[\|\tilde{x}\|^2] = d$ and

$$\mathbb{E}_{\tilde{x} \sim \mathcal{N}(0, I_d)}[\sigma(\langle w, x \rangle)\sigma(\langle v, x \rangle)] = \frac{\sin\phi(w, v) + (\pi - \phi(w, v))\cos\phi(w, v)}{2\pi}\|w\|\|v\|$$

(see Cho & Saul (2009) or Safran & Shamir (2018)). Moreover, the first equality follows from that fact that for $\tilde{x} \sim \mathcal{N}(0, I_d)$, $r^2 := \|\tilde{x}\|^2$ and $\phi := \tilde{x}/\|\tilde{x}\|$ are random variables that independently follow the Chi-squared distribution and the uniform distribution on $\mathbb{S}^{d-1}$ respectively, and therefore,

$$\mathbb{E}_{\tilde{x} \sim \mathcal{N}(0, I_d)}[\sigma(\langle w, x \rangle)\sigma(\langle v, x \rangle)] = \mathbb{E}_{\tilde{x} \sim \mathcal{N}(0, I_d)}[r^2\sigma(\langle w, \phi \rangle)\sigma(\langle v, \phi \rangle)]$$

$$= \mathbb{E}_{x \sim P_X}[\sigma(\langle w, x \rangle)\sigma(\langle v, x \rangle)] \cdot \mathbb{E}_{\tilde{x} \sim \mathcal{N}(0, I_d)}[\|\tilde{x}\|^2].$$

By using Eq. (2), we get

$$\mathcal{R}_\lambda(\theta) = \frac{1}{2}\mathbb{E}_x\left[\left(f_{a^{\circ}, W^{\circ}}(x) - f(x; \bar{\theta})\right)^2\right] + \lambda\|\theta\|^2$$

$$= \frac{1}{2}\mathbb{E}_x\left[\left(f_{a^{\circ}, W^{\circ}}(x)\right)^2\right] - \sum_{i,j=1}^{m} \bar{a}_i a_j^{\circ} I(\bar{w}_i, w_j^{\circ}) + \frac{1}{2}\sum_{i,j=1}^{m} \bar{a}_i \bar{a}_j I(\bar{w}_i, \bar{w}_j) + \lambda\|\theta\|^2,$$

where $\bar{w}$ is the element-wise clipping of $w \in \mathbb{R}^d$ and

$$I(w, v) = \frac{\sin \phi(w, v) + (\pi - \phi(w, v)) \cos \phi(w, v)}{2\pi d} \|w\| \|v\|.$$

Next, we move to derive the gradient of $\mathcal{R}_\lambda(\cdot)$. Note that $\frac{\mathrm{d}\bar{r}}{\mathrm{d}r} = |r|/\cosh^2(r|r|/2R)$. Then, since $e^r + e^{-r} \geq 2 + |r|$ for $r \in \mathbb{R}$, we have that $\cosh(r|r|/2R) \geq 1 + r^2/4R$, and hence $|\frac{\mathrm{d}\bar{r}}{\mathrm{d}r}| \leq \frac{|r|}{(1+r^2/4R)^2} \leq \min\{|r|, 16R^2|r|/r^4\} \leq 4R$. Moreover, through a straightforward calculation, we can show that $\frac{\mathrm{d}\bar{r}}{\mathrm{d}r}$ is 1-Lipschitz (in other words, the mapping $r \mapsto \bar{r}$ is 1-smooth).

Using this, each component of the gradient of $\mathcal{R}_\lambda(\cdot)$ can be written as follows:

$$\nabla_{a_j} \mathcal{R}_\lambda(\theta) = \sum_{i=1}^m \bar{a}_i I(\bar{w}_i, \bar{w}_j) \cdot \frac{\mathrm{d}\bar{a}_j}{\mathrm{d}a_j} - \sum_{i=1}^m a_i^\circ I(w_i^\circ, \bar{w}_j) \cdot \frac{\mathrm{d}\bar{a}_j}{\mathrm{d}a_j} + 2\lambda a_j \tag{3}$$

$$\nabla_{w_j} \mathcal{R}_\lambda(\theta) = -\sum_{i=1}^m \bar{a}_i a_j^\circ J(\bar{w}_i, w_j^\circ) \odot \frac{\mathrm{d}\bar{w}_j}{\mathrm{d}w_j} + \sum_{i=1}^m \bar{a}_i \bar{a}_j J(\bar{w}_i, \bar{w}_j) \odot \frac{\mathrm{d}\bar{w}_j}{\mathrm{d}w_j} + 2\lambda w_j, \tag{4}$$

where $\odot$ denotes the Hadamard product and

$$J(w, v) = \frac{\|v\| \|w\|^{-1} \sin \phi(w, v) w + (\pi - \phi(w, v)) v}{2\pi d},$$

which is the gradient of $I(w, v)$ with respect to $w$ (see Brutzkus & Globerson (2017) or Safran & Shamir (2018)).

## C  PROOF OF PROPOSITION 4.3

This section provides the convergence guarantee for phase I. Our objective is to give the proof of Proposition 4.3. To this end, we first introduce the theory around the gradient Langevin dynamics exploited in Vempala & Wibisono (2019).

### C.1  A BRIEF NOTE ON THE GRADIENT LANGEVIN DYNAMICS

In their analysis, the following notion of the *log-Sobolev inequality* plays the essential role, which defined as follows:

**Definition C.1.** *A probability distribution with a density function $q$ satisfies the **log-Sobolev inequality (LSI)** if there exists a constant $\alpha > 0$ such that for all smooth function $g$, it holds that*

$$\mathbb{E}_q[g^2 \log g^2] - \mathbb{E}_q[g^2] \log \mathbb{E}_q[g^2] \leq \frac{2}{\alpha} \mathbb{E}_q[\|\nabla g\|^2].$$

*$\alpha$ is called a log-Sobolev constant.*

It is known that the LSI is equivalent to the following inequality:

$$H_q(p) \leq \frac{1}{2\alpha} J_q(p) \quad (\forall p \in \mathcal{P}), \tag{5}$$

where $H_q(p) := \int_\mathbb{R} p(\theta) \log \frac{p(\theta)}{q(\theta)} \mathrm{d}x$ is the KL divergence, $J_q(p) := \int_\mathbb{R} p(\theta) \left\| \nabla \log \frac{p(\theta)}{q(\theta)} \right\|^2 \mathrm{d}\theta$ is the relative Fisher information, and $\mathcal{P}$ is the set of all probability density functions.

Now we consider the sampling from the probability distribution $q$ over $\mathbb{R}^d$. We assume that $-\log q(\cdot) : \mathbb{R}^d \to \mathbb{R}$ is differentiable. One of the well-known and promising approaches is updating the parameter $\theta^{(0)}$ sampled from an initial distribution $\rho_0$ as follows:

$$\theta^{(k+1)} = \theta^{(k)} - \eta \nabla(-\log q)(\theta^{(k)}) + \sqrt{2\eta} \zeta^{(k)}, \tag{6}$$

where $\eta > 0$ is a constant and $\zeta^{(k)} \sim \mathcal{N}(0, I_d)$ is an independent standard Gaussian random variable. Vempala & Wibisono (2019) shows that if the LSI holds and $-\log q$ has a smoothness, the sufficient number of updates (6) actually achieves the sampling from $q$, in a sense that the KL divergence between the distribution of $\theta^{(k)}$ and $q$ will be small.

**Theorem C.2** ((Vempala & Wibisono, 2019, Theorem 1)). *Suppose that a probability measure with a density function $q$ satisfies the LSI and $-\log q$ is $L$-smooth. Then for any $\theta^{(0)} \sim p_0$ with $H_q(p_0)$, the sequence $(\theta^{(k)})_{k=0}^{\infty}$ with step-size $0 < \eta < \frac{\alpha}{4L^2}$ satisfies*

$$H_q(p_t) \leq \exp(-\alpha\eta k)H_q(p_0) + \frac{8\eta dL^2}{\alpha}.$$

*Hence for any $\delta > 0$, the output of the update* (6) *with step-size $\eta \leq \frac{\alpha}{4L^2}\min\{1, \frac{\delta}{4d}\}$ achieves $H_q(p_t) < \delta$ after $k \geq \frac{1}{\alpha\eta}\log\frac{2H_q(p_t)}{\delta}$ iterations.*

## C.2 PROOF OF LEMMA 4.2

The goal of this section is to prove Proposition 4.3, the convergence of gradient Langevin dynamics. As we stated in Section 4.2, we consider the value of $\mathcal{R}_\lambda(\cdot)$ instead of $\widehat{\mathcal{R}}_\lambda(\cdot)$, and ensure its value will decrease enough. To this end, we first prove Lemma 4.2, which evaluates the difference between $\nabla\mathcal{R}_\lambda(\cdot)$ and $\nabla\widehat{\mathcal{R}}_\lambda(\cdot)$.

*proof of Lemma 4.2.* The proof of Lemma 4.2 is basically based on that of Theorem 1 in Mei et al. (2018a) and Lemma 5.3 in Zhang et al. (2019). For the notational simplicity we denote $m(d + 1) =: D$. Let $N_\epsilon$ be the $\epsilon$-covering number of $\mathbf{B}\left(0, \sqrt{D}R\right)$ with respect to the $\ell_2$-distance. Let $\Theta_\epsilon = \{\theta_1, \ldots, \bar{\theta}_N\}$ be a corresponding $\epsilon$-cover with $|\Theta_\epsilon| = N$. It is known that $\log N = D\log\left(3\sqrt{D}R/\epsilon\right)$ is sufficient to ensure the existence of such covering.

First we note that $\nabla\mathcal{R}_\lambda(\theta) - \nabla\widehat{\mathcal{R}}_\lambda(\theta) = \frac{1}{n}\sum_{i=1}^n \nabla\ell(y_i, f(x_i; \bar{\theta})) - \nabla\mathbb{E}[\ell(y, f(x; \bar{\theta}))]$. For each $\theta \in \Theta$, let $j(\theta) \in \arg\min_{j\in[N]}\|\bar{\theta} - \bar{\theta}_j\|$ and $\widehat{\theta} := \bar{\theta}_{j(\theta)}$. For $\theta \in \mathbf{B}\left(0, \sqrt{D}R\right)$, we consider the following decomposition:

$$\frac{1}{n}\sum_{i=1}^n \nabla\ell(y_i, f(x_i; \bar{\theta})) - \nabla\mathbb{E}[\ell(y, f(x; \bar{\theta}))] = \frac{1}{n}\sum_{i=1}^n\Big[\nabla\ell(y_i, f(x_i; \bar{\theta})) - \nabla\ell(y_i, f(x_i; \widehat{\theta}))\Big]$$
$$+ \left(\frac{1}{n}\sum_{i=1}^n \nabla\ell(y_i, f(x_i; \widehat{\theta})) - \nabla\mathbb{E}[\ell(y, f(x; \widehat{\theta}))]\right)$$
$$+ \Big(\nabla\mathbb{E}[\ell(y, f(x; \widehat{\theta}))] - \nabla\mathbb{E}[\ell(y, f(x; \bar{\theta}))]\Big).$$

This gives that

$$\left\|\frac{1}{n}\sum_{i=1}^n \nabla\ell(y_i, f(x_i; \bar{\theta})) - \nabla\mathbb{E}[\ell(y, f(x; \bar{\theta}))]\right\| \leq \left\|\frac{1}{n}\sum_{i=1}^n\Big[\nabla\ell(y_i, f(x_i; \bar{\theta})) - \nabla\ell(y_i, f(x_i; \widehat{\theta}))\Big]\right\|$$
$$+ \left\|\frac{1}{n}\sum_{i=1}^n \nabla\ell(y_i, f(x_i; \widehat{\theta})) - \nabla\mathbb{E}[\ell(y, f(x; \widehat{\theta}))]\right\|$$
$$+ \left\|\nabla\mathbb{E}[\ell(y, f(x; \widehat{\theta}))] - \nabla\mathbb{E}[\ell(y, f(x; \bar{\theta}))]\right\|,$$

and hence it holds that

$$
\mathrm{P}\left(\sup_{\theta\in\mathbf{B}\left(0,\sqrt{D}R\right)}\left\|\frac{1}{n}\sum_{i=1}^{n}\nabla\ell(y_i,f(x_i;\bar{\theta}))-\nabla\mathbb{E}[\ell(y,f(x;\bar{\theta}))]\right\|\geq t\right)
$$

$$
\leq\underbrace{\mathrm{P}\left(\sup_{\theta\in\mathbf{B}\left(0,\sqrt{D}R\right)}\left\|\frac{1}{n}\sum_{i=1}^{n}\Big[\nabla\ell(y_i,f(x_i;\bar{\theta}))-\nabla\ell(y_i,f(x_i;\widehat{\theta}))\Big]\right\|\geq\frac{t}{3}\right)}_{(\mathrm{I})}
$$

$$
+\underbrace{\mathrm{P}\left(\sup_{\theta\in\mathbf{B}\left(0,\sqrt{D}R\right)}\left\|\frac{1}{n}\sum_{i=1}^{n}\nabla\ell(y_i,f(x_i;\widehat{\theta}))-\nabla\mathbb{E}[\ell(y,f(x;\widehat{\theta}))]\right\|\geq\frac{t}{3}\right)}_{(\mathrm{II})}
$$

$$
+\underbrace{\mathrm{P}\left(\sup_{\theta\in\mathbf{B}\left(0,\sqrt{D}R\right)}\left\|\nabla\mathbb{E}[\ell(y,f(x;\widehat{\theta}))]-\nabla\mathbb{E}[\ell(y,f(x;\bar{\theta}))]\right\|\geq\frac{t}{3}\right)}_{(\mathrm{III})}
$$

for any $t>0$. Then we evaluate the each term of the RHS.

**Upper bound on (I):** Since $\nabla\ell(y_i,f(x_i;\bar{\theta}))=2\big(f(x_i;\bar{\theta})-y_i\big)\nabla f(x_i;\bar{\theta})$, it holds that

$$
\nabla\ell(y_i,f(x_i;\bar{\theta}))-\nabla\ell(y_i,f(x_i;\widehat{\theta}))
$$
$$
=2\Big(f(x_i;\bar{\theta})-f(x_i;\widehat{\theta})\Big)\nabla f(x_i;\bar{\theta})-2(f(x_i;\widehat{\theta})-y_i)(\nabla f(x_i;\widehat{\theta})-\nabla f(x_i;\bar{\theta})).
$$

Therefore, we have that

$$
\mathrm{P}\left(\sup_{\theta\in\mathbf{B}\left(0,\sqrt{D}R\right)}\left\|\frac{1}{n}\sum_{i=1}^{n}\Big[\nabla\ell(y_i,f(x_i;\bar{\theta}))-\nabla\ell(y_i,f(x_i;\widehat{\theta}))\Big]\right\|\geq\frac{t}{3}\right)
$$

$$
\leq\mathrm{P}\left(\sup_{\theta\in\mathbf{B}\left(0,\sqrt{D}R\right)}\left\|\frac{2}{n}\sum_{i=1}^{n}\Big[\Big(f(x_i;\bar{\theta})-f(x_i;\widehat{\theta})\Big)\nabla f(x_i;\bar{\theta})\Big]\right\|\geq\frac{t}{6}\right)
$$

$$
+\mathrm{P}\left(\sup_{\theta\in\mathbf{B}\left(0,\sqrt{D}R\right)}\left\|\frac{2}{n}\sum_{i=1}^{n}\Big[(f(x_i;\widehat{\theta})-y_i)(\nabla f(x_i;\widehat{\theta})-\nabla f(x_i;\bar{\theta}))\Big]\right\|\geq\frac{t}{6}\right),
$$

Since the mapping $\widehat{\theta}\mapsto f(x;\widehat{\theta})$ is $2R$-Lipschitz and $\left\|\nabla f(x;\bar{\theta})\right\|\leq 2mR$ for any $\theta\in\Theta$, it holds that the first term must be zero as long as $t\geq 4mR^2\epsilon$. As for the second term, since $|f(x;\bar{\theta})-y_i|\leq mR^2+U+1$ for any $x_i,y_i$ and $\theta\in\Theta$, it holds that

$$
(\mathrm{I})=\mathrm{P}\left(\sup_{\theta\in\mathbf{B}\left(0,\sqrt{D}R\right)}\left\|\frac{2}{n}\sum_{i=1}^{n}\Big[(f(x_i;\widehat{\theta})-y_i)(\nabla f(x_i;\widehat{\theta})-\nabla f(x_i;\bar{\theta}))\Big]\right\|\geq\frac{t}{6}\right)
$$

$$
\leq\mathrm{P}\left(\sup_{\theta\in\mathbf{B}\left(0,\sqrt{D}R\right)}\left\|\frac{2}{n}\sum_{i=1}^{n}\Big[\nabla f(x_i;\widehat{\theta})-\nabla f(x_i;\bar{\theta})\Big]\right\|\geq\frac{t}{6(mR^2+U+1)}\right).
$$

Hence, we move to evaluate $\sup_{\theta\in\mathbf{B}\left(0,\sqrt{D}R\right)}\left\|\frac{2}{n}\sum_{i=1}^{n}\Big[\nabla f(x_i;\widehat{\theta})-\nabla f(x_i;\bar{\theta})\Big]\right\|$. To this end, we consider the decomposition

$$
\left\|\frac{2}{n}\sum_{i=1}^{n}\Big[\nabla f(x_i;\widehat{\theta})-\nabla f(x_i;\bar{\theta})\Big]\right\|
$$

$$
\leq\sum_{j=1}^{m}\left(\left\|\frac{2}{n}\sum_{i=1}^{n}\Big[\nabla_{a_j}f(x_i;\widehat{\theta})-\nabla_{a_j}f(x_i;\bar{\theta})\Big]\right\|+\left\|\frac{2}{n}\sum_{i=1}^{n}\Big[\nabla_{w_j}f(x_i;\widehat{\theta})-\nabla_{w_j}f(x_i;\bar{\theta})\Big]\right\|\right),
$$

where

$$\nabla_{a_j} f(x_i; \bar{\theta}) = \sigma(\langle \bar{w}_j, x_i \rangle) \frac{\mathrm{d}\bar{a}_j}{\mathrm{d}a_j}, \qquad \nabla_{w_j} f(x_i; \bar{\theta}) = \bar{a}_j \mathbb{1}\{\langle \bar{w}_j, x_i \rangle \geq 0\} x_i \odot \frac{\mathrm{d}\bar{w}_j}{\mathrm{d}w_j}.$$

This decomposition implies that

$$(\mathrm{I}) \leq \mathrm{P}\left( \max_{j \in [m]} \sup_{\theta \in \mathbf{B}(0, \sqrt{D}R)} \left\| \frac{2}{n} \sum_{i=1}^{n} \left[ \nabla_{a_j} f(x_i; \hat{\theta}) - \nabla_{a_j} f(x_i; \bar{\theta}) \right] \right\| \geq \frac{t}{12m(mR^2 + U + 1)} \right)$$

$$+ \mathrm{P}\left( \max_{j \in [m]} \sup_{\theta \in \mathbf{B}(0, \sqrt{D}R)} \left\| \frac{2}{n} \sum_{i=1}^{n} \left[ \nabla_{w_j} f(x_i; \hat{\theta}) - \nabla_{w_j} f(x_i; \bar{\theta}) \right] \right\| \geq \frac{t}{12m(mR^2 + U + 1)} \right).$$

For each term, it holds that

$$\left\| \nabla_{a_j} f(x_i; \hat{\theta}) - \nabla_{a_j} f(x_i; \bar{\theta}) \right\| \leq \left\| (\sigma(\langle \bar{w}_j, x_i \rangle) - \sigma(\langle \hat{w}_j, x_i \rangle)) \frac{\mathrm{d}\bar{a}_j}{\mathrm{d}a_j} \right\| + \left\| \sigma(\langle \hat{w}_j, x_i \rangle) \left( \frac{\mathrm{d}\bar{a}_j}{\mathrm{d}a_j} - \frac{\mathrm{d}\hat{a}_j}{\mathrm{d}a_j} \right) \right\|$$

$$\leq \| \bar{w}_j - \hat{w}_j \| \left| \frac{\mathrm{d}\bar{a}_j}{\mathrm{d}a_j} \right| + \left| \frac{\mathrm{d}\bar{a}_j}{\mathrm{d}a_j} - \frac{\mathrm{d}\hat{a}_j}{\mathrm{d}a_j} \right|$$

$$\leq 4R \| \bar{w}_j - \hat{w}_j \| + 2|\bar{a}_j - \hat{a}_j| \leq 4R\epsilon + \epsilon$$

and

$$\left\| \nabla_{w_j} f(x_i; \hat{\theta}) - \nabla_{w_j} f(x_i; \bar{\theta}) \right\| \leq \left\| \bar{a}_j (\mathbb{1}\{\langle \bar{w}_j, x_i \rangle \geq 0\} - \mathbb{1}\{\langle \hat{w}_j, x_i \rangle \geq 0\}) x_i \odot \frac{\mathrm{d}\bar{w}_j}{\mathrm{d}w_j} \right\|$$

$$+ \left\| (\bar{a}_j - \hat{a}_j) \mathbb{1}\{\langle \hat{w}_j, x_i \rangle \geq 0\} x_i \odot \frac{\mathrm{d}\bar{w}_j}{\mathrm{d}w_j} \right\|$$

$$+ \left\| \hat{a}_j \mathbb{1}\{\langle \bar{w}_j, x_i \rangle \geq 0\} x_i \odot \left( \frac{\mathrm{d}\bar{w}_j}{\mathrm{d}w_j} - \frac{\mathrm{d}\hat{w}_j}{\mathrm{d}w_j} \right) \right\|$$

$$\leq R \left\| (\mathbb{1}\{\langle \bar{w}_j, x_i \rangle \geq 0\} - \mathbb{1}\{\langle \hat{w}_j, x_i \rangle \geq 0\}) x_i \odot \frac{\mathrm{d}\bar{w}_j}{\mathrm{d}w_j} \right\|$$

$$+ 4R \| \bar{a}_j - \hat{a}_j \| + 4R \| \bar{w}_j - \hat{w}_j \|$$

$$\leq R \left\| (\mathbb{1}\{\langle \bar{w}_j, x_i \rangle \geq 0\} - \mathbb{1}\{\langle \hat{w}_j, x_i \rangle \geq 0\}) x_i \odot \frac{\mathrm{d}\bar{w}_j}{\mathrm{d}w_j} \right\| + 8R\epsilon.$$

The first term can be bounded by

$$\left\| (\mathbb{1}\{\langle \bar{w}_j, x_i \rangle \geq 0\} - \mathbb{1}\{\langle \hat{w}_j, x_i \rangle \geq 0\}) x_i \odot \frac{\mathrm{d}\bar{w}_j}{\mathrm{d}w_j} \right\| \leq \| (\mathbb{1}\{\langle \bar{w}_j, x_i \rangle \geq 0\} - \mathbb{1}\{\langle \hat{w}_j, x_i \rangle \geq 0\}) x_i \| \cdot \| \hat{w}_j \|$$

$$\leq \mathbb{1}\{|\langle \hat{w}_j, x_i \rangle| \leq \epsilon\} \cdot \| \hat{w}_j \|,$$

where the last inequality follows from $|\langle \bar{w}_j, x_i \rangle - \langle \hat{w}_j, x_i \rangle| \leq \| \bar{w}_j - \hat{w}_j \| \cdot \| x_i \| \leq \epsilon$. Therefore, we obtain that

$$(\mathrm{I}) \leq \mathrm{P}\left( \max_{j \in [m]} \sup_{\theta \in \mathbf{B}(0, \sqrt{D}R)} \frac{\#\{i \in [n] \mid |\langle \hat{w}_j, x_i \rangle| \leq \epsilon\} \cdot \| \hat{w}_j \|}{n} \geq \frac{t}{24mR(mR^2 + U + 1)} \right)$$

$$= \mathrm{P}\left( \max_{\hat{\theta} \in \Theta_\epsilon, j \in [m]} \frac{\#\{i \in [n] \mid |\langle \hat{w}_j, x_i \rangle| \leq \epsilon\} \cdot \| \hat{w}_j \|}{n} \geq \frac{t}{24mR(mR^2 + U + 1)} \right)$$

as long as $\frac{t}{24mR(mR^2 + U + 1)} \geq \max\{4R\epsilon, \epsilon, 8R\epsilon\} = 8R\epsilon$. We have that

$$\mathrm{P}\left( \frac{\#\{i \in [n] \mid |\langle \hat{w}_j, x_i \rangle| \leq \epsilon\} \cdot \| \hat{w}_j \|}{n} \geq \frac{t}{24mR(mR^2 + U + 1)} \right)$$

$$= \mathrm{P}\left( \frac{\#\{i \in [n] \mid |\langle \hat{w}_j, x_i \rangle| \leq \epsilon\}}{n} \geq \frac{t}{24mR(mR^2 + U + 1)\| \hat{w}_j \|} \right)$$

when $\hat{w}_j \neq \mathbf{0}$. If $\hat{w}_j = \mathbf{0}$, the LHS must be zero as long as $t > 0$. Lemma 12 in Cai et al. (2013) shows that for each $j$ and $i$, the angle between $\hat{w}_j$ and $x_i$ is distributed with density function

$$h(\phi) = \frac{1}{\sqrt{\pi}} \frac{\Gamma(\frac{d}{2})}{\Gamma(\frac{d-1}{2})} \cdot (\sin \phi)^{d-2} : \qquad \phi \in [0, \pi].$$

Since $\left| \frac{\pi}{2} - \phi \right| \leq \Delta$ implies $|\cos \phi| = \left| \sin\left( \frac{\pi}{2} - \phi \right) \right| \leq \Delta$ for any $\Delta > 0$ and $h(\phi) \leq \frac{1}{\sqrt{\pi}} \frac{\Gamma(\frac{d}{2})}{\Gamma(\frac{d-1}{2})}$ for any $\phi \in [0, \pi]$, it holds that

$$P(|\langle \hat{w}_j, x_i \rangle| \leq \epsilon) \leq P\left( \left| \frac{\pi}{2} - \phi_{ij} \right| \leq \frac{\epsilon}{\|\hat{w}_j\|} \right) \leq \frac{2\epsilon}{\|\hat{w}_j\|} \frac{1}{\sqrt{\pi}} \frac{\Gamma(\frac{d}{2})}{\Gamma(\frac{d-1}{2})} \leq \frac{2\sqrt{d}\epsilon}{\sqrt{\pi}\|\hat{w}_j\|},$$

where $\phi_{ij}$ is the angle between $\hat{w}_j$ and $x_i$. Therefore, $\#\{i \in [n] \mid |\langle \hat{w}_j, x_i \rangle| \leq \epsilon\}$ follows the Binomial distribution $B(n, \mathsf{p})$ with $\mathsf{p} \leq \frac{2\sqrt{d}\epsilon}{\sqrt{\pi}\|\hat{w}_j\|}$. Since a random variables that follows the Binomial distribution is bounded and especially sub-Gaussian Wainwright (2019), it holds that

$$P\left( \frac{\#\{i \in [n] \mid |\langle \hat{w}_j, x_i \rangle| \leq \epsilon\}}{n} \geq s + \frac{2\sqrt{d}\epsilon}{\sqrt{\pi}\|\hat{w}_j\|} \right) \leq P\left( \frac{\#\{i \in [n] \mid |\langle \hat{w}_j, x_i \rangle| \leq \epsilon\}}{n} \geq s + \mathsf{p} \right)$$

$$\leq \exp(-2ns^2).$$

for an arbitrarily $s > 0$. By taking uniform bound, we obtain that

$$P\left( \max_{\hat{\theta} \in \Theta_\epsilon, j \in [m]} \frac{\#\{i \in [n] \mid |\langle \hat{w}_j, x_i \rangle| \leq \epsilon\}}{n} \geq s + \frac{2\sqrt{d}\epsilon}{\sqrt{\pi}\|\hat{w}_j\|} \right) \leq N \exp(-2ns^2)$$

Hence, as long as $\epsilon \leq \frac{\sqrt{\pi}}{96mR(mR^2+U+1)2\sqrt{d}} t$ (verified later in this proof), by letting $s = \frac{t}{48mR(mR^2+U+1)\|\hat{w}_j\|}$, we obtain that

$$P\left( \max_{\hat{\theta} \in \Theta_\epsilon, j \in [m]} \frac{\#\{i \in [n] \mid |\langle \hat{w}_j, x_i \rangle| \leq \epsilon\}}{n} \geq s + \frac{d\epsilon}{\sqrt{\pi}\|\hat{w}_j\|} \right)$$

$$\leq mN \exp\left( -2n\left( \frac{t}{24mR(mR^2+U+1)\|\hat{w}_j\|} \right)^2 \right)$$

$$\leq mN \exp\left( -\frac{2n}{dR^2}\left( \frac{t}{24mR(mR^2+U+1)} \right)^2 \right),$$

where the last inequality follows from $\|\hat{w}_j\|^2 \leq dR^2$. As a result, the term (I) can be bounded by

$$\text{(I)} \leq mN \exp\left( -\frac{2n}{dR^2}\left( \frac{t}{24mR(mR^2+U+1)} \right)^2 \right).$$

**Upper bound on (II):** First, we observe that the term (II) is equivalent to

$$P\left( \max_{j \in [N]} \left\| \frac{1}{n} \sum_{i=1}^n \nabla\ell(y_i, f(x_i; \bar{\theta}_j)) - \nabla\mathbb{E}[\ell(y, f(x; \bar{\theta}_j))] \right\| \geq \frac{t}{3} \right).$$

For each $j \in [N]$, a straightforward calculation gives that $\|\nabla\ell(y_i, f(x_i; \theta_j))\| \leq 2R(mR^2+1)$, and hence the vector $\nabla\ell(y_i, f(x_i; \theta_j))$ is sub-Gaussian with a parameter $R(mR^2+1)$, i.e., it holds that

$$P\left( \left\| \frac{1}{n} \sum_{i=1}^n \nabla\ell(y_i, f(x_i; \bar{\theta}_j)) - \nabla\mathbb{E}[\ell(y, f(x; \bar{\theta}_j))] \right\| \geq \frac{t}{3} \right) \leq 2e^{-\frac{nt^2}{18G^2}}$$

with $G = R(mR^2+1)$ for arbitrary $t > 0$. By taking uniform bound, we obtain

$$P\left( \max_{j \in [N]} \left\| \frac{1}{n} \sum_{i=1}^n \nabla\ell(y_i, f(x_i; \bar{\theta}_j)) - \nabla\mathbb{E}[\ell(y, f(x; \bar{\theta}_j))] \right\| \geq \frac{t}{3} \right) \leq 2Ne^{-\frac{t^2}{18G^2}}.$$

**Upper bound on (III):** The goal is obtaining (III) = 0 for a sufficiently small $\epsilon$. Particularly, we assume that $\epsilon < 1$ here. To this end, we aim to show

$$\left\| \nabla \mathbb{E}[\ell(y, f(x; \widehat{\theta}))] - \nabla \mathbb{E}[\ell(y, f(x; \bar{\theta}))] \right\| \leq cL'\epsilon^{1/2} \tag{7}$$

with a constant $c > 0$ and $L = O(m^2 R^3)$. First we consider the case where the absolute value of the each component in $\bar{\theta}$ is bounded by $1/2$. By Lemma C.5, it holds that

$$\left\| \nabla \mathbb{E}[\ell(y, f(x; \widehat{\theta}))] - \nabla \mathbb{E}[\ell(y, f(x; \bar{\theta}))] \right\| \leq L' \|\theta - \theta_{j(\theta)}\| = L' \cdot \left( \|\theta - \theta_{j(\theta)}\|^2 \right)^{\frac{1}{2}}$$

for any $\theta \in \Theta$ with $L' = O(m^2 R^3)$. Moreover, a straightforward calculation shows that a mapping $r \mapsto 2R \tanh^{-1}(r/R)$ (the inverse mapping of $r \mapsto R \tanh(r/2R)$) is 8-Lipschitz in $[0, 1/2]$, we have $\|\theta - \theta_{j(\theta)}\|^2 \leq 8\|\bar{\theta} - \widehat{\theta}\| \leq 8\epsilon$. Therefore, we obtain that

$$\left\| \nabla \mathbb{E}[\ell(y, f(x; \widehat{\theta}))] - \nabla \mathbb{E}[\ell(y, f(x; \bar{\theta}))] \right\| \leq L'(8\epsilon)^{1/2},$$

i.e., Eq. (7) with $c = 8$. Assume that there is a component of $\theta$ whose absolute value is greater than $1/2$. First, suppose that a component of $\bar{w}_j$ is greater than $1/2$ for $j \in [m]$. We consider the decomposition

$$\left\| \nabla_{w_j} \mathbb{E}[\ell(y, f(x; \widehat{\theta}))] - \nabla_{w_j} \mathbb{E}[\ell(y, f(x; \bar{\theta}))] \right\|$$

$$\leq \left\| \nabla_{\widehat{w}_j} \mathbb{E}[\ell(y, f(x; \widehat{\theta}))] - \nabla_{\bar{w}_j} \mathbb{E}[\ell(y, f(x; \bar{\theta}))] \right\| \cdot \left\| \frac{\mathrm{d}\widehat{w}_j}{\mathrm{d}w_j} \right\| + \left\| \nabla_{\bar{w}_j} \mathbb{E}[\ell(y, f(x; \bar{\theta}))] \right\| \cdot \left\| \frac{\mathrm{d}\widehat{w}_j}{\mathrm{d}w_j} - \frac{\mathrm{d}\bar{w}_j}{\mathrm{d}w_j} \right\|.$$

Since $\|\bar{w}_j\| > 1/2$, we can check that the mapping $\widehat{w}_j \mapsto \mathbb{E}[\ell(y, f(x; \bar{\theta}))]$ is $L''$ smooth with $L'' = O(mR^2 d^{-1/2})$ according to its Hessian (see Safran & Shamir (2018)). Since $\left\| \frac{\mathrm{d}\widehat{w}_j}{\mathrm{d}w_j} \right\| \leq 4\sqrt{d}R$, the first term is at most $O(mR^3) \cdot \epsilon$. Since $\left\| \mathbb{E}[\ell(y, f(x; \bar{\theta}))] \right\| \leq 2R(mR^2 + 1)$ and $r \mapsto \bar{r}$ is 1-smooth, the second term is at most $O(mR^3) \cdot \epsilon$. Hence we get that $\left\| \nabla_{w_j} \mathbb{E}[\ell(y, f(x; \widehat{\theta}))] - \nabla_{w_j} \mathbb{E}[\ell(y, f(x; \bar{\theta}))] \right\| \leq O(mR^3) \cdot \epsilon$. In the case $|a_j| > 1/2$ for $j \in [m]$, the same bound also holds with $\left\| \nabla_{a_j} \mathbb{E}[\ell(y, f(x; \widehat{\theta}))] - \nabla_{a_j} \mathbb{E}[\ell(y, f(x; \bar{\theta}))] \right\|$. By using these bound instead of Lemma E.5 and $\epsilon < 1$, we obtain the same bound Eq. (7) in this case. Eq. (7) implies (III) = 0 as long as $\frac{t}{3} \geq cL'\epsilon^{1/2}$, which gives the assertion.

**Combining (I)–(III):** Combining these bounds, we get that

$$P\left( \sup_{\theta \in \mathbf{B}(0, \sqrt{D}R)} \left\| \frac{1}{n} \sum_{i=1}^{n} \nabla \ell(y_i, f(x_i; \bar{\theta})) - \nabla \mathbb{E}[\ell(y, f(x; \bar{\theta}))] \right\| \geq t \right)$$

$$\leq mN \exp\left( -\frac{2n}{dR^2} \left( \frac{t}{24mR(mR^2 + U + 1)} \right)^2 \right) + 2Ne^{-\frac{t^2}{18G^2}} + 0$$

$$= \exp\left( D \log \frac{3\sqrt{D}R}{\epsilon} \right)$$

$$\cdot \left[ m \exp\left( -\frac{2n}{dR^2} \left( \frac{t}{24mR(mR^2 + U + 1)} \right)^2 \right) + 2 \exp\left( -\frac{nt^2}{18R^2(mR^2 + U + 1)^2} \right) \right]$$

as long as $t \geq C_0 \max\{mR^2\epsilon, mR(mR^2 + U)\epsilon, L'\epsilon^{1/2}\}$ holds with a constant $C_0 > 0$. By letting $t = C_1 L'\epsilon^{1/2}$ and $\epsilon = C_2 \frac{d \log \delta}{nm^2}$ with constants $C_1 > 0$ and $C_2 > 0$, we obtain the conclusion. $\square$

## C.3 PROOF OF THE CONVERGENCE IN PHASE I

Based on the results so far, we move to the proof of Proposition 4.3. The proof is conducted in two-step. First, we evaluate the "distance" between the $\pi_\infty$ and the distribution of $\theta^{(k)}$. Moreover, it is

ensured that the function value $\mathcal{R}_\lambda(\theta)$, where $\theta$ is sampled from $\pi_\infty$, will be small for a sufficiently large $\beta$. Combining these two facts, we can guarantee that the function value $\mathcal{R}_\lambda(\theta^{(k)})$ also will be small, which concludes Proposition 4.3. The following proposition ensures the convergence of the marginal distribution of $\theta^{(k)}$ to the invariant measure $\pi_\infty$:

**Proposition C.3.** *Suppose that the probability measure $\pi_\infty$ satisfies the LSI with a constant $\alpha$ and $\mathcal{R}_\lambda(\cdot)$ is $L$-smooth with $L > 1$. Let $q$ be a density function of $\pi_\infty$ (i.e., $q(\theta) \propto \exp(-\beta\mathcal{R}_\lambda(\theta))$) with $\beta > 2$. For any $\theta^{(0)} \sim \rho_0$ with $H_q(\rho_0) < +\infty$, the sequence $(\theta^{(k)})_{k=0}^\infty$ with step-size $0 < \eta^{(1)} < \frac{\alpha}{4\beta L^2}$ satisfies*

$$H_q(\rho_k) \leq \exp\left(-\frac{\alpha\eta^{(1)}}{\beta}k\right)H_q(\rho_0) + \frac{16\beta\eta^{(1)}DL^2}{\alpha} + \frac{32\beta V_{grad}^2}{3\alpha},$$

*where $D := m(d+1)$, $\rho_k$ is the density function of the marginal distribution of $\theta^{(k)}$, and $V_{grad}$ is a constant introduced in Lemma 4.2. In particular, for any $\delta > 0$, the output of phase I with step-size $\eta^{(1)} \leq \frac{\delta\alpha}{32\beta L^2 D}$ achieves $H_q(\rho_k) < \delta + \frac{32\beta V_{grad}^2}{3\alpha}$ after $k \geq \frac{\beta}{\alpha\eta^{(1)}}\log\frac{2H_q(\rho_0)}{\delta}$ iterations.*

As we stated in Section 4.2, our result extends the existing one Vempala & Wibisono (2019) in the sense that it gives the convergence for the non-differential objective function $\widehat{\mathcal{R}}_\lambda(\cdot)$. Indeed, this difference appears in the last term, $\frac{32\beta V_{grad}^2}{3\alpha}$. Since $V_{grad}^2 \lesssim n^{-1}$ by Lemma 4.2, we can ensure that this error diverges to zero as the sample size $n$ increases. To apply this result to ensure the convergence of the phase I, we just need to check that the invariant measure $\pi_\infty$ satisfies the LSI and $\mathcal{R}_\lambda$ is smooth, and we clarify them as follows:

**Lemma C.4** (log-Sobolev inequality)**.** *The invariant measure $\pi_\infty$ satisfies the LSI with a constant $\alpha = 2\beta\lambda\exp(-8\beta m^2 R^4)$.*

**Lemma C.5** (smoothness)**.** *$\mathcal{R}_\lambda(\cdot)$ is $L$-smooth, i.e., for any $\theta$, $\theta' \in \Theta$, $\left\|\nabla\mathcal{R}_\lambda(\theta) - \nabla\mathcal{R}_\lambda(\theta')\right\| \leq L\left\|\theta - \theta'\right\|$ holds with $L = O(m^2 R^3 + \lambda)$.*

The proof of these lemmas can be seen in Appendix E.

**Remark C.6.** *In Lemma C.4, the LSI constant $\alpha$ depends exponentially on $m$, which results in the exponential dependency of phase I convergence on $m$. This is caused by the fact that the sup-norm of the student network depends on $m$ (see the proof of Lemma C.4). Therefore, we can indeed remove this dependency by considering the following settings: (1) utilizing the mean field network (multiplying $1/m$ to the output of the student), (2) $w_j$s are directed to different directions to some extent, and hence the sup-norm can be bounded.*

To ensure Proposition C.3, we first show the following lemma, which evaluates the each step of the gradient Langevin dynamics.

**Lemma C.7.** *Suppose that $\pi_\infty$ satisfies the LSI with a constant $\alpha$ and $\mathcal{R}_\lambda(\cdot)$ is $L$-smooth with $L > 1$, and $\beta > 2$. Then for any $\theta^{(0)} \sim \rho_0$ with $H_q(\rho_0) < +\infty$, if $0 < \eta < \frac{\alpha}{4\beta L^2}$, it holds that*

$$H_q(\rho_{k+1}) \leq e^{-\alpha\eta/\beta}H_q(\rho_k) + 12\eta^2 DL^2 + 8\eta V_{grad}^2,$$

*where $\rho_k$ is the density function of the marginal distribution of $\theta^{(k)}$ and $V_{grad}$ is the constant defined in Lemma 4.2.*

*Proof.* The proof of Lemma C.7 is basically based on that of Lemma 3 in Vempala & Wibisono (2019). For notational simplicity suppose $k = 0$ and let $\theta_0 = \theta^{(0)}$. The one step of the gradient Langevin dynamics

$$\theta^{(1)} = \theta^{(0)} - \eta\nabla\widehat{\mathcal{R}}_\lambda\left(\theta^{(0)}\right) + \sqrt{\frac{2\eta}{\beta}}\zeta^{(0)}$$

can be seen as an output at time $\eta\beta^{-1}$ of the following SDE:

$$\mathrm{d}\theta_t = -\beta\nabla\widehat{\mathcal{R}}_\lambda(\theta_0)\mathrm{d}t + \sqrt{2}\mathrm{d}B_t,$$

where $\{B_t\}_{t \geq 0}$ is the standard Brownian motion in $\Theta$ ($= \mathbb{R}^{(d+1) \times m}$). As Vempala & Wibisono (2019), it holds that

$$\frac{\partial \rho_{t|0}(\theta_t|\theta_0)}{\partial t} = \nabla \cdot \left( \rho_{t|0}(\theta_t|\theta_0) \beta \nabla \widehat{\mathcal{R}}_\lambda(\theta_0) \right) + \Delta \rho_{t|0}(\theta_t|\theta_0),$$

and therefore,

$$\frac{\mathrm{d}}{\mathrm{d}t} H_q(\rho_t) = -J_q(\rho_t) + \beta \cdot \mathbb{E}_{\rho_{0t}} \left[ \left\langle \nabla \mathcal{R}_\lambda(\theta_t) - \nabla \widehat{\mathcal{R}}_\lambda(\theta_0), \nabla \log \frac{\rho_t(\theta_t)}{q(\theta_t)} \right\rangle \right],$$

where $\rho_{t|0}(\cdot|\theta_0)$ the conditional density, and $\rho_{t0}$ is the density of the joint distribution of $\theta_0$ and $\theta_t$.

Then we evaluate the second term. The inner product in this term can be bounded by

$$\left\langle \nabla \mathcal{R}_\lambda(\theta_t) - \nabla \widehat{\mathcal{R}}_\lambda(\theta_0), \nabla \log \frac{\rho_t(\theta_t)}{q(\theta_t)} \right\rangle \leq \left\| \nabla \mathcal{R}_\lambda(\theta_t) - \nabla \widehat{\mathcal{R}}_\lambda(\theta_0) \right\|^2 + \frac{1}{4} \left\| \nabla \log \frac{\rho_t(\theta_t)}{q(\theta_t)} \right\|^2$$

$$\leq 2 \left\| \nabla \mathcal{R}_\lambda(\theta_t) - \nabla \mathcal{R}_\lambda(\theta_0) \right\|^2$$

$$+ 2 \left\| \nabla \mathcal{R}_\lambda(\theta_0) - \nabla \widehat{\mathcal{R}}_\lambda(\theta_0) \right\|^2 + \frac{1}{4} \left\| \nabla \log \frac{\rho_t(\theta_t)}{q(\theta_t)} \right\|^2.$$

In the above bound, we use $\langle a, b \rangle \leq a^2 + b^2/4$ for $a, b \in \mathbb{R}^D$ in the first inequality and $\|a - b\|^2 \leq 2\|a\|^2 + 2\|b\|^2$ for $a, b \in \mathbb{R}^D$ in the second inequality. Therefore, by using Lemma 4.2, we get that

$$\mathbb{E}_{\rho_{0t}} \left[ \left\langle \nabla \mathcal{R}_\lambda(\theta_t) - \nabla \mathcal{R}_\lambda(\theta_0), \nabla \log \frac{\rho_t(\theta_t)}{q(\theta_t)} \right\rangle \right]$$

$$\leq 2V_{grad}^2 + 2\mathbb{E}_{\rho_{0t}} \left[ \left\| \nabla \mathcal{R}_\lambda(\theta_t) - \nabla \mathcal{R}_\lambda(\theta_0) \right\|^2 \right] + \frac{1}{4} \mathbb{E}_{\rho_{0t}} \left[ \left\| \nabla \log \frac{\rho_t(\theta_t)}{q(\theta)} \right\|^2 \right]$$

$$= 2V_{grad}^2 + 2\mathbb{E}_{\rho_{0t}} \left[ \left\| \nabla \mathcal{R}_\lambda(\theta_t) - \nabla \mathcal{R}_\lambda(\theta_0) \right\|^2 \right] + \frac{1}{4} J_q(\rho_t).$$

Then the second term is bounded by

$$\mathbb{E}_{\rho_{0t}} \left[ \left\| \nabla \mathcal{R}_\lambda(\theta_t) - \nabla \mathcal{R}_\lambda(\theta_0) \right\|^2 \right] \leq L^2 \mathbb{E}_{\rho_{0t}} \left[ \|\theta_t - \theta_0\|^2 \right]$$

$$= L^2 \mathbb{E}_{\rho_{0t}} \left[ \left\| -t \nabla \widehat{\mathcal{R}}_\lambda(\theta_0) + \sqrt{\frac{2t}{\beta}} \zeta^{(0)} \right\|^2 \right]$$

$$= t^2 L^2 \mathbb{E}_{\rho_{0t}} \left[ \left\| \nabla \mathcal{R}_\lambda(\theta_0) + (\nabla \widehat{\mathcal{R}}_\lambda(\theta_0) - \nabla \mathcal{R}_\lambda(\theta_0)) \right\|^2 \right]$$

$$+ L^2 \mathbb{E}_{\rho_{0t}} \left[ \left\| \sqrt{\frac{2t}{\beta}} \zeta^{(0)} \right\|^2 \right]$$

$$\leq 2t^2 L^2 \left( \mathbb{E}_{\rho_{0t}} \left[ \|\nabla \mathcal{R}_\lambda(\theta_0)\|^2 \right] + V_{grad}^2 \right) + L^2 \frac{2t}{\beta} D$$

$$\leq \frac{1}{\beta} \left( \frac{4t^2 L^4}{\alpha} H_q(\rho_0) + 2t^2 L^3 D \right) + 2\eta^2 L^2 V_{grad}^2 + t L^2 D.$$

In the last inequality, we use Lemma 10 in Vempala & Wibisono (2019) and $\beta > 2$. Thus we obtain

$$\frac{\mathrm{d}}{\mathrm{d}t} H_q(\rho_t) \leq -\frac{3}{4} J_q(\rho_t) + \frac{8\beta t^2 L^4}{\alpha} H_q(\rho_0) + 4\beta t^2 L^3 D + 2\beta t L^2 D + (2\beta \eta^2 L^2 + 2) V_{grad}^2$$

$$\leq -\frac{3\alpha}{2} H_q(\rho_t) + \frac{8\beta t^2 L^4}{\alpha} H_q(\rho_0) + 6\beta t L^2 D + 4\beta V_{grad}^2$$

since the LSI (Eq. (5)) holds and $tL \leq \eta L \leq 1$. Multiplying both sides by $e^{3\alpha t/2}$ and integrating them from $t = 0$ to $t = \eta \beta^{-1}$, we get

$$e^{3\alpha \eta / 2\beta} H_q(\rho_\eta) - H_q(\rho_0) \leq \frac{2(e^{3\alpha \eta / 2\beta} - 1)}{3\alpha} \left( \frac{4\beta \eta^2 L^4}{\alpha} H_q(\rho_0) + 6\beta \eta D L^2 + 4\beta V_{grad}^2 \right)$$

$$\leq 2\eta \left( \frac{8\eta^2 L^4}{\alpha} H_q(\rho_0) + 6\eta D L^2 + 4V_{grad}^2 \right),$$

where we use the inequality $e^a \leq 1 + 2a$ for $a \in [0, 1]$ and $3\alpha\eta/2\beta \leq 1$ (derived from the assumption of $\eta$). Rearranging this inequality, we have

$$H_q(\rho_\eta) \leq e^{-3\alpha\eta/2\beta}\left(1 + \frac{16\eta^3 L^4}{\alpha}\right)H_q(\rho_0) + e^{-3\alpha\eta/2\beta}\left(12\eta^2 DL^2 + 8V_{grad}^2\eta\right)$$

$$\leq e^{-\alpha\eta/\beta}H_q(\rho_0) + 12\eta^2 DL^2 + 8\eta V_{grad}^2,$$

where the last inequality follows from $1 + \frac{16\eta^3 L^4}{\alpha} \leq 1 + \frac{\alpha\eta}{16\beta^2} \leq 1 + \frac{\alpha\eta}{2\beta} \leq e^{\alpha\eta/2\beta}$. By replacing $\rho_0$ by $\rho_k$ and $\rho_\eta$ by $\rho_{k+1}$, we get the conclusion. $\qquad\square$

*proof of Proposition C.3.* By Lemma C.7, it holds that

$$H_q(\rho_k) \leq e^{-\alpha\eta k/\beta}H_q(\rho_0) + \left(12\eta^2 DL^2 + 8\eta V_{grad}^2\right)\sum_{k'=1}^{k} e^{-\alpha\eta k'/\beta}$$

$$\leq e^{-\alpha\eta k/\beta}H_q(\rho_0) + \frac{12\eta^2 DL^2 + 8\eta V_{grad}^2}{1 - e^{-\alpha\eta/\beta}} \leq e^{-\alpha\eta k/\beta}H_q(\rho_0) + \frac{16\beta\eta DL^2}{\alpha} + \frac{32\beta V_{grad}^2}{3\alpha},$$

where, the last inequality follows from $L > 1$ (derived from Lemma C.5) and $1 - e^{-c} \geq \frac{3}{4}c$ for $c \in [0, \frac{1}{4}]$ and $\frac{\alpha\eta}{\beta} < \frac{1}{4L^2} < \frac{1}{4}$. Thus we get the assertion. $\qquad\square$

*proof of Proposition 4.3.* By the Otto-Villani theorem, it holds that $\mathcal{W}_2(\rho_k, q)^2 \leq \frac{2}{\alpha}H_q(\rho_k)$. Therefore, Proposition C.3 implies that after $k \geq \frac{\beta}{\alpha\eta}\log\frac{2H_q(\rho_0)}{\delta}$ iteration, it holds that

$$\mathcal{W}_2(\rho_k, q) \leq \sqrt{\frac{2}{\alpha}\left(\delta + \frac{32\beta V_{grad}^2}{3\alpha}\right)}$$

Then we obtain that

$$\mathbb{E}[\mathcal{R}_\lambda(\theta^{(k)})] - \mathcal{R}_\lambda^* \leq \left(\mathbb{E}[\mathcal{R}_\lambda(\theta^{(k)})] - \mathbb{E}_{\pi_\infty}[\mathcal{R}_\lambda(\theta)]\right) + \left(\mathbb{E}_{\pi_\infty}[\mathcal{R}_\lambda(\theta)] - \mathcal{R}_\lambda^*\right)$$

$$\leq C(\lambda + m)\sqrt{\frac{2}{\alpha}\left(\delta + \frac{32\beta V_{grad}^2}{3\alpha}\right)} + \frac{D}{2\beta}\log\left(\frac{eL}{M}\left(\frac{b\beta}{D} + 1\right)\right),$$

where we use Lemma E.1 and Lemma E.2 for the inequality. By specifying $\alpha$, $L$, $M$ and $b$ by applying Lemma C.4, Lemma C.5, and Lemma E.3, we get the conslusion. $\qquad\square$

## D    PROOF OF THEOREM 4.6

The objective of this section is to prove Theorem 4.6. First, by the noisy gradient descent, the objective value decreases enough, and we can ensure that for each node of the teacher network, there exists a node of the student network that is "close" to each other. Then we can prove the local convergence property based on the strong convexity around the parameters of the teacher network.

The proof of the local convergence relies on that of Zhang et al. (2019). They consider the setting where the parameters of the second layer are all positive, i.e., $a_j = a_j^\circ = 1$ for all $j \in [m]$ and provide the following proposition:

**Proposition D.1** (Theorem 4.2 of Zhang et al. (2019)). *Let $f^\circ : x \mapsto \sum_{j=1}^{m} \sigma(\langle w_j^\circ, x\rangle)$ be a teacher network with parameters $W^\circ = (w_1^\circ \ w_2^\circ \ \cdots \ w_m^\circ) \in \mathbb{R}^{d \times m}$, $\kappa = \sigma_1/\sigma_m$ is the condition number of $W^\circ$, and $\sigma = (\prod_{j=1}^{m}\sigma_j)/\sigma_m^m$. Assume the inputs $(x_i)_{i=1}^{n}$ are sampled from $\mathcal{N}(0, I_d)$, and the outputs $(y_i)_{i=1}^{n}$ are generated from the teacher network. Suppose that the initial estimator $W^{(0)}$ satisfies $\|W^{(0)} - W^\circ\|_F \leq c\sigma_m/\kappa^3 m^2$, where $c > 0$ is a small enough absolute constant. Then there exists absolute constants $c_1$, $c_2$, $c_3$, $c_4$, and $c_5$ such that under*

$$n \geq \frac{c_1\kappa^{10}m^9 d}{\sigma_m}\log\left(\frac{\kappa md}{\sigma_m}\right) \cdot \left(\|W^*\|_F^2 + v^2\right),$$

*the output of the gradient descent with step-size $\eta \leq \frac{1}{c_2 \kappa m^2}$ satisfies*

$$\|W^{(k)} - W^\circ\|_F^2 \leq \left(1 - \frac{c_3 \eta}{\sigma \kappa^2}\right)^k \|W^{(0)} - W^\circ\|_F^2 + \frac{c_4 \sigma^2 \kappa^4 m^5 d \log n}{n} \cdot \left(\|W^\circ\|_F^2 + v^2\right)$$

*with probability at least $1 - c_5 d^{-10}$.*

Their proof can also be applied to the setting in this paper, i.e., $a_j$, $a_j^\circ \in \{\pm 1\}$ holds, and if a teacher node $j$ and a student node $k_j$ are close to each other, it holds that $a_j^\circ = a_{k_j}$. In Proposition 4.3, if it holds that $\mathbb{E}[\mathcal{R}_\lambda(\theta^{(k)})] - \mathcal{R}_\lambda^* \leq b\epsilon_0$, we obtain that

$$\mathcal{R}_\lambda(\theta^{(k)}) - \mathcal{R}_\lambda^* \leq \epsilon_0 \tag{8}$$

with probability at least $1 - b$, by using the Markov inequality. In the rest of this section, we assume that (8) is satisfied. In this case, if Proposition 4.4 is ensured, we can apply Proposition D.1 and Theorem 4.6 is proved. We give its proof in the rest of this section.

*proof of Proposition 4.4.* Let $\theta^\circ = ((a_1^\circ, w_1^\circ), \ldots, (a_m^\circ, w_m^\circ))$. Then by $\mathcal{R}_\lambda(\theta) - \mathcal{R}_\lambda(\theta^\circ) \leq \mathcal{R}_\lambda(\theta) - \mathcal{R}_\lambda^* \leq \epsilon_0$, it holds that

$$\frac{1}{2}\mathbb{E}_x\left[(f_{a^\circ, W^\circ}(x) - f(x; \theta))^2\right] + \lambda \sum_{j=1}^m \left(|a_j|^2 + \|w_j\|^2\right) \leq \lambda \sum_{j=1}^m \left(|a_j^\circ|^2 + \|w_j^\circ\|^2\right) + \epsilon_0,$$

and therefore,

$$\frac{1}{2}\mathbb{E}_x\left[(f_{a^\circ, W^\circ}(x) - f(x; \theta))^2\right] \leq \frac{\epsilon_0}{m} \sum_{j=1}^m \left(|a_j^\circ|^2 + \|w_j^\circ\|^2\right) + \epsilon_0$$

$$\leq \frac{\epsilon_0}{m} \sum_{j=1}^m \left(1 + \|W^\circ\|_F^2\right) + \epsilon_0 \leq 3\epsilon_0, \tag{9}$$

where we use $|a_j^\circ|^2 = 1$ for all $j \in [m]$ and $\sum_{j=1}^m \|w_j^\circ\|^2 = \|W^\circ\|_F^2 \leq m\|W^\circ\|_2^2 \leq m$. Then we move to evaluate the LHS. Since $\sigma(u) = \frac{u + |u|}{2}$ for $u \in \mathbb{R}$, it holds that

$$f(x; \theta) = \sum_{j=1}^m a_j \sigma(\langle w_j, x\rangle) = \frac{1}{2} \sum_{j=1}^m a_j(|\langle w_j, x\rangle| + \langle w_j, x\rangle),$$

$$f_{a^\circ, W^\circ}(x) = \sum_{j=1}^m a_j \sigma(\langle w_j^\circ, x\rangle) = \frac{1}{2} \sum_{j=1}^m a_j^\circ\left(|\langle w_j^\circ, x\rangle| + \langle w_j^\circ, x\rangle\right)$$

Hence we have that

$$\frac{1}{2}\mathbb{E}_x\left[(f_{a^\circ, W^\circ}(x) - f(x; \theta))^2\right]$$

$$= \frac{1}{8}\mathbb{E}_x\left[\left(\sum_{j=1}^m a_j^\circ\left(|\langle w_j^\circ, x\rangle| + \langle w_j^\circ, x\rangle\right) - \sum_{j=1}^m a_j(|\langle w_j, x\rangle| + \langle w_j, x\rangle)\right)^2\right]$$

$$= \frac{1}{8}\mathbb{E}_x\left[\left(\sum_{j=1}^m a_j^\circ|\langle w_j^\circ, x\rangle| - \sum_{j=1}^m a_j|\langle w_j, x\rangle|\right)^2\right] + \frac{1}{8}\mathbb{E}_x\left[\left\langle\sum_{j=1}^m a_j^\circ w_j^\circ - \sum_{j=1}^m a_j w_j, x\right\rangle^2\right],$$

where the last equality follows from $\mathbb{E}_x[|\langle w_1, x\rangle|\langle w_2, x\rangle] = 0$ for all $w_1, w_2 \in \mathbb{R}^d$, which follows from the fact that the distribution $P_X$ is symmetric. Then Eq. (9) gives that

$$\mathbb{E}_x\left[\left(\sum_{j=1}^m a_j^\circ|\langle w_j^\circ, x\rangle| - \sum_{j=1}^m a_j|\langle w_j, x\rangle|\right)^2\right] \leq 24\epsilon_0, \tag{10}$$

$$\mathbb{E}_x\left[\left\langle\sum_{j=1}^m a_j^\circ w_j^\circ - \sum_{j=1}^m a_j w_j, x\right\rangle^2\right] \leq 24\epsilon_0. \tag{11}$$

The analysis based on Eq. (10), the error analysis of student networks with the absolute value activation, is conducted in Zhou et al. (2021). Here we import Lemma D.2 from their technique. They focus on the setting where $a_j^\circ = 1$ for all $j \in [m]$, but we can apply it here. Then we get that for every $j \in [m]$, there exists $k_j \in [m]$ and a constant $C > 0$ such that $\arccos\big(|\langle w_j^\circ, w_k\rangle|/\|w_j^\circ\|\|w_k\|\big) \leq Cm\sigma_{\min}^{-5/3}\epsilon^{1/3}$ and $\big\||a_{k_j}|w_{k_j} - w_j^\circ\big\| \leq \mathrm{poly}(m, \sigma_{\min}^{-1})\epsilon^{3/8}$.

We simply denote $k_j$ by $j$. Since Zhou et al. (2021) uses the absolute value for the activation, it may hold that $\arccos\big(\langle w_j^\circ, w_j\rangle/\|w_j^\circ\|\|w_j\|\big) > \pi/2$ (i.e., $w_j^\circ$ and $w_k$ have "opposite" directions). From now on, we omit such cases by Eq. (11). Let $\mathbf{a} = (a_1, \ldots, a_m) = (a_1^\circ, \ldots, a_m^\circ)$ and $W_\Delta = (w_1^\circ - w_1, \ldots, w_m^\circ - w_m)$. And we denote the angle between $w_j^\circ$ and $w_j$ by $\phi_j$. Then, Eq. (11) can be rewritten as

$$\mathbb{E}_{x \sim P_X}\big[(\mathbf{a}^\mathsf{T} W_\Delta x)^2\big] \leq 24\epsilon_0.$$

Let $\tilde{x} \sim \mathcal{N}(0, I_d)$, since $r^2 := \|\tilde{x}\|^2$ and $\phi := \tilde{x}/\|\tilde{x}\|$ are random variables that independently follow the Chi-squared distribution and the uniform distribution on $\mathbb{S}^{d-1}$ respectively. Hence it holds that

$$\mathbb{E}_{x \sim P_X}\big[(\mathbf{a}^\mathsf{T} W_\Delta x)^2\big] = \frac{\mathbb{E}_{\tilde{x} \sim \mathcal{N}(0, I_d)}\big[(\mathbf{a}^\mathsf{T} W_\Delta \tilde{x})^2\big]}{\mathbb{E}_{\tilde{x} \sim \mathcal{N}(0, I_d)}\|\tilde{x}\|^2} = \frac{\mathbb{E}_{r \sim \mathcal{N}(0, \|\langle \mathbf{a}, W_\Delta\rangle\|^2)}\big[r^2\big]}{d} = \frac{\|\langle \mathbf{a}, W_\Delta\rangle\|^2}{d} \leq 24\epsilon_0.$$

This implies $\|\langle \mathbf{a}, W_\Delta\rangle\|^2 \leq 24\epsilon_0 d$. Since $w_j^\circ - w_j = (1 - \langle w_j^\circ, w_j\rangle)w_j^\circ + (\langle w_j^\circ, w_j\rangle w_j^\circ - w_j)$ and $\|\langle w_j^\circ, w_j\rangle w_j^\circ - w_j\| = \sin\phi_j$, we have that

$$\begin{aligned}
\langle \mathbf{a}, W_\Delta\rangle =& \Big((1 - \langle w_1^\circ, w_1\rangle)a_1, \ldots, (1 - \langle w_m^\circ, w_m\rangle)a_m\Big)^\mathsf{T} W^\circ \\
& + \left(\langle \mathbf{a}, W_\Delta\rangle - \Big((1 - \langle w_1^\circ, w_1\rangle)a_1, \ldots, (1 - \langle w_m^\circ, w_m\rangle)a_m\Big)^\mathsf{T} W^\circ\right) \\
=& \Big((1 - \langle w_1^\circ, w_1\rangle)a_1, \ldots, (1 - \langle w_m^\circ, w_m\rangle)a_m\Big)^\mathsf{T} W^\circ \\
& + \Big\langle \mathbf{a}, \big(\langle w_1^\circ, w_1\rangle w_1^\circ - w_1, \ldots, \langle w_m^\circ, w_m\rangle w_m^\circ - w_m\big)\Big\rangle
\end{aligned}$$

and the second term is at most $O(m^{3/2}\sigma_{\min}^{-5/3}\epsilon^{1/3})$. As for the first term, it holds that it is at least $\sigma_{\min} \sum_{j=1}^m (1 - \langle w_j^\circ, w_j\rangle)^2$. Hence, by letting $\epsilon = o(d^{-1}m^{-3/2}\sigma_{\min}^8)$, it must hold that $\langle w_j^\circ, w_j\rangle > 0$, which gives the assertion. $\qquad\square$

**Lemma D.2** (Lemma 9 and Lemma 10 in Zhou et al. (2021)). *Assume that $x \sim \mathcal{N}(0, I_d)$ and $f^\circ : x \mapsto \sum_{j=1}^m |\langle w_j^\circ, x\rangle|$ is a teacher network with parameters $w_1^\circ, \ldots, w_m^\circ \in \mathbb{R}^d$ satisfying $\min_{j_1, j_2} \arccos\big(\langle w_{j_1}^\circ, w_{j_2}^\circ\rangle/\|w_{j_1}^\circ\|\|w_{j_2}^\circ\|\big) \geq \Delta$ for $\Delta > 0$ and $0 < w_{\min} \leq \|w_j^\circ\| \leq w_{\max}$ for all $j \in [m]$. Then there exists a threshold $\epsilon_0 = \mathrm{poly}(\Delta, m^{-1}, w_{\max}.w_{\min})$ such that if a student network $\widehat{f} : x \mapsto \sum_{j=1}^m |\langle w_j, x\rangle|$ satisfies $\mathbb{E}_x[(f^\circ - \widehat{f})^2] \leq \epsilon \leq \epsilon_0$, it holds that for every $j \in [m]$, there exists $k_j \in [m]$ and a constant $C > 0$ such that $\arccos\big(|\langle w_j^\circ, w_k\rangle|/\|w_j^\circ\|\|w_k\|\big) \leq Cmw_{\max}w_{\min}^{-5/3}\epsilon^{1/3}$ and $\big\||a_{k_j}|w_{k_j} - w_j^\circ\big\| \leq \mathrm{poly}(m, \Delta^{-1}, w_{\max})\epsilon^{3/8}$.*

## E  AUXILIARY LEMMAS

### E.1  EVALUATION OF THE INVARIANT MEASURE

This subsection provides lemmas about the evaluation of the function value sampled from the invariant measure $\beta$. These are utilized in the proof of Proposition 4.3 (see Appendix C). First, we introduce two results from Raginsky et al. (2017), and then we prove the *dissipativity*, which is imposed as an assumption in these results.

**Lemma E.1** (Proposition 11 in Raginsky et al. (2017)). *Suppose that $f : \Theta \to \mathbb{R}$ satisfies the following conditions:*

- *$f$ is L-smooth.*

- $f$ is $(M, b)$-dissipative, i.e., it holds that $\langle \theta, \nabla f(\theta) \rangle \geq M\|\theta\|^2 - b$ for any $\theta \in \Theta$.

Then, for any $\beta \geq 2/M$, it holds that

$$\mathbb{E}_{\theta \sim \pi_\infty}[f(\theta)] - \min_{\theta \in \Theta} f(\theta) \leq \frac{d}{2\beta} \log\left(\frac{eL}{M}\left(\frac{b\beta}{d} + 1\right)\right)$$

**Lemma E.2** (Lemma 2 and Lemma 6 in Raginsky et al. (2017)). *Let $\mu_1$, $\mu_2$ be two probability measures on $\Theta$ with finite second moments, and let $f : \Theta \to \mathbb{R}$ be a $(M, b)$-dissipative function satisfying $\|\nabla f(0)\| \leq B$ for $B \geq 0$. Then, it holds that*

$$\left| \int_\Theta g \mathrm{d}\mu_1 - \int_\Theta g \mathrm{d}\mu_2 \right| \leq (M\sigma + B)\mathcal{W}_2(\mu_1, \mu_2),$$

*where $\sigma^2 := \max\{\int_\Theta \|\theta\|^2 \mathrm{d}\mu_1, \int_\Theta \|\theta\|^2 \mathrm{d}\mu_2\}$.*

**Lemma E.3** (dissipativity). *$\mathcal{R}_\lambda(\cdot)$ is $(M, b)$-dissipative with $M = 2\lambda$ and $b = 8m^2R^3$.*

*Proof.* By a straightforward calculation, we have that

$$\langle \theta, \nabla \mathcal{R}_\lambda(\theta) \rangle = \sum_{j=1}^m a_j \nabla_{a_j} \mathcal{R}_\lambda(\theta) + \sum_{j=1}^m \langle w_j, \nabla_{w_j} \mathcal{R}_\lambda(\theta) \rangle$$

$$= \sum_{j=1}^m a_j \left[ \sum_{i=1}^m \bar{a}_i I(\bar{w}_i, \bar{w}_j) \cdot \frac{\mathrm{d}\bar{a}_j}{\mathrm{d}a_j} - \sum_{i=1}^m a_i^\circ I(w_i^\circ, \bar{w}_j) \cdot \frac{\mathrm{d}\bar{a}_j}{\mathrm{d}a_j} + 2\lambda a_j \right]$$

$$+ \sum_{j=1}^m \left\langle w_j, -\sum_{i=1}^m \bar{a}_i a_j^\circ J(\bar{w}_i, w_j^\circ) \odot \frac{\mathrm{d}\bar{w}_j}{\mathrm{d}w_j} + \sum_{i=1}^m \bar{a}_i \bar{a}_j J(\bar{w}_i, \bar{w}_j) \odot \frac{\mathrm{d}\bar{w}_j}{\mathrm{d}w_j} + 2\lambda w_j \right\rangle$$

$$= 2\lambda \|\theta\|^2 + \sum_{j=1}^m a_j \left[ \sum_{i=1}^m \bar{a}_i I(\bar{w}_i, \bar{w}_j) \cdot \frac{\mathrm{d}\bar{a}_j}{\mathrm{d}a_j} - \sum_{i=1}^m a_i^\circ I(w_i^\circ, \bar{w}_j) \cdot \frac{\mathrm{d}\bar{a}_j}{\mathrm{d}a_j} \right]$$

$$+ \sum_{j=1}^m \left\langle w_j, -\sum_{i=1}^m \bar{a}_i a_j^\circ J(\bar{w}_i, w_j^\circ) \odot \frac{\mathrm{d}\bar{w}_j}{\mathrm{d}w_j} + \sum_{i=1}^m \bar{a}_i \bar{a}_j J(\bar{w}_i, \bar{w}_j) \odot \frac{\mathrm{d}\bar{w}_j}{\mathrm{d}w_j} \right\rangle.$$

As for the second term and the third term, since $|I(w, v)| \leq \|w\|\|v\|/2d$ and $\|J(w, v)\| \leq \|v\|/2d$ for any $w, v \in \mathbb{R}^d$, we have that

$$\left| \sum_{j=1}^m a_j \left[ \sum_{i=1}^m \bar{a}_i I(\bar{w}_i, \bar{w}_j) \cdot \frac{\mathrm{d}\bar{a}_j}{\mathrm{d}a_j} - \sum_{i=1}^m a_i^\circ I(w_i^\circ, \bar{w}_j) \cdot \frac{\mathrm{d}\bar{a}_j}{\mathrm{d}a_j} \right] \right|$$

$$\leq \sum_{j=1}^m \left| a_j \frac{\mathrm{d}\bar{a}_j}{\mathrm{d}a_j} \left[ \sum_{i=1}^m \bar{a}_i I(\bar{w}_i, \bar{w}_j) - \sum_{i=1}^m a_i^\circ I(w_i^\circ, \bar{w}_j) \right] \right| \leq 4m^2R^3,$$

and

$$\left| \sum_{j=1}^m \left\langle w_j, -\sum_{i=1}^m \bar{a}_i a_j^\circ J(\bar{w}_i, w_j^\circ) \odot \frac{\mathrm{d}\bar{w}_j}{\mathrm{d}w_j} + \sum_{i=1}^m \bar{a}_i \bar{a}_j J(\bar{w}_i, \bar{w}_j) \odot \frac{\mathrm{d}\bar{w}_j}{\mathrm{d}w_j} \right\rangle \right|$$

$$\leq \sum_{j=1}^m \left| \left\langle w_j, -\sum_{i=1}^m \bar{a}_i a_j^\circ J(\bar{w}_i, w_j^\circ) \odot \frac{\mathrm{d}\bar{w}_j}{\mathrm{d}w_j} + \sum_{i=1}^m \bar{a}_i \bar{a}_j J(\bar{w}_i, \bar{w}_j) \odot \frac{\mathrm{d}\bar{w}_j}{\mathrm{d}w_j} \right\rangle \right| \leq 4m^2R^3.$$

Combining these inequality, we get that

$$\langle \theta, \nabla \mathcal{R}_\lambda(\theta) \rangle \geq 2\lambda \|\theta\|^2 - 8m^2R^3,$$

which gives the conclusion. $\square$

### E.2 PROOF OF LEMMA C.4

In this subsection, we give a proof to Lemma C.4, the LSI for the invariant measure $\pi_\infty$. The key notion is that $\mathcal{R}_\lambda$ can be decomposed to the bounded term ($L_2$-distance) and the strongly convex term (regularization term). Combining this fact with the following lemma, we can ensure the LSI.

**Lemma E.4** (Holley & Stroock (1987); Nitanda et al. (2021b)). *Let a probability measure on $\Theta$ with a density function $q$ satisfy the LSI with a constant $\alpha$. For a function $f : \Theta \to \mathbb{R}$ that satisfies $|f(\theta)| \leq B$ for any $\theta \in \Theta$, a probability measure defined by*

$$Q(\theta)\mathrm{d}\theta := \frac{\exp(f(\theta))q(\theta)}{\mathbb{E}_q[\exp(f(\theta))q(\theta)]}\mathrm{d}\theta$$

*satisfies the LSI with a constant $\alpha \exp(-4B)$.*

*proof of Lemma C.4.* First, we note that

$$\exp(-\beta \mathcal{R}_\lambda(\theta))\mathrm{d}\theta = \exp\Big(-\beta\lambda\|\theta\|^2\Big) \cdot \exp\Big(-\frac{\beta}{2}\mathbb{E}_x\Big[\big(f_{a^\circ,W^\circ}(x) - f(x;\bar{\theta})\big)^2\Big]\Big)\mathrm{d}\theta.$$

Since the function $\theta \mapsto \beta\lambda\|\theta\|^2$ is $2\beta\lambda$-strongly convex, a measure with density $\exp\Big(-\beta\lambda\|\theta\|^2\Big)\mathrm{d}\theta$ satisfies the LSI with a constant $\beta\lambda$ Bakry & Émery (1985). Moreover, by a straightforward calculation shows that $\frac{\beta}{2}\mathbb{E}_x\Big[\big(f_{a^\circ,W^\circ}(x) - f(x;\bar{\theta})\big)^2\Big] \leq 2\beta m^2 R^4$, Lemma E.4 implies that $\pi_\infty$ satisfies the LSI with a constant $2\beta\lambda\exp\big(-8\beta m^2 R^4\big)$, which gives the conclusion. $\square$

### E.3 PROOF OF LEMMA C.5

In this subsection we write $\mathcal{L}(\theta) := \frac{1}{2}\mathbb{E}_x\Big[\big(f_{a^\circ,W^\circ}(x) - f(x;\bar{\theta})\big)^2\Big]$, i.e., $\mathcal{R}_\lambda(\theta) := \mathcal{L}(\theta) + \lambda\|\theta\|^2$. Since $\theta \mapsto \lambda\|\theta\|^2$ is $2\lambda$-smooth, it is sufficient to show that $\mathcal{L}(\cdot)$ is $L'$-smooth with $L' = O(m^2 R^3)$ for proving Lemma C.5. To this end, let $\theta, \theta' \in \Theta$. We consider the decomposition

$$\big\|\nabla\mathcal{L}(\theta) - \nabla\mathcal{L}(\theta')\big\| = \sqrt{\sum_{j=1}^m\Big(\big|\nabla_{a_j}\mathcal{L}(\theta) - \nabla_{a_j}\mathcal{L}(\theta')\big|^2 + \big\|\nabla_{w_j}\mathcal{L}(\theta) - \nabla_{w_j}\mathcal{L}(\theta')\big\|^2\Big)}$$

$$\leq \sum_{j=1}^m\big(\big|\nabla_{a_j}\mathcal{L}(\theta) - \nabla_{a_j}\mathcal{L}(\theta')\big| + \big\|\nabla_{w_j}\mathcal{L}(\theta) - \nabla_{w_j}\mathcal{L}(\theta')\big\|\big),$$

where

$$\nabla_{a_j}\mathcal{L}(\theta) - \nabla_{a_j}\mathcal{L}(\theta') = \sum_{i=1}^m\Big(\bar{a}_i I(\bar{w}_i, \bar{w}_j) \cdot \frac{\mathrm{d}\bar{a}_j}{\mathrm{d}a_j} - \bar{a}'_i I(\bar{w}'_i, \bar{w}'_j) \cdot \frac{\mathrm{d}\bar{a}'_j}{\mathrm{d}a_j}\Big)$$

$$- \sum_{i=1}^m\Big(a_i^\circ I(w_i^\circ, \bar{w}_j) \cdot \frac{\mathrm{d}\bar{a}_j}{\mathrm{d}a_j} - a_i^\circ I(w_i^\circ, \bar{w}'_j) \cdot \frac{\mathrm{d}\bar{a}'_j}{\mathrm{d}a_j}\Big),$$

$$\nabla_{w_j}\mathcal{L}(\theta) - \nabla_{w_j}\mathcal{L}(\theta') = -\sum_{i=1}^m\Big(\bar{a}_i a_j^\circ J(\bar{w}_i, w_j^\circ) \odot \frac{\mathrm{d}\bar{w}_j}{\mathrm{d}w_j} - \bar{a}'_i a_j^\circ J(\bar{w}'_i, w_j^\circ) \odot \frac{\mathrm{d}\bar{w}'_j}{\mathrm{d}w_j}\Big)$$

$$+ \frac{1}{2}\sum_{i=1}^m\Big(\bar{a}_i \bar{a}_j J(\bar{w}_i, \bar{w}_j) \odot \frac{\mathrm{d}\bar{w}_j}{\mathrm{d}w_j} - \bar{a}'_i \bar{a}'_j J(\bar{w}'_i, \bar{w}'_j) \odot \frac{\mathrm{d}\bar{w}'_j}{\mathrm{d}w_j}\Big).$$

(see Eq. (3) and Eq. (4)). The following lemma gives an upper bound for each term.

**Lemma E.5.** *For any $\theta, \theta' \in \Theta$ and $j \in [m]$, it holds that*

$$\big|\nabla_{a_j}\mathcal{L}(\theta) - \nabla_{a_j}\mathcal{L}(\theta')\big| \leq m\Big(\frac{5R^3}{2} + \frac{2\sqrt{d}R}{d}\Big)\big(|a_j - a'_j| + \|w_j - w'_j\|\big) + 2R^3\sum_{i=1}^m\|\bar{w}_i - \bar{w}'_i\|$$

$$\big\|\nabla_{w_j}\mathcal{L}(\theta) - \nabla_{w_j}\mathcal{L}(\theta')\big\| \leq m\Big(\frac{2R}{d} + 2R^3\Big)\big(|a_j - a'_j| + \|w_j - w'_j\|\big) + 2R^3\sum_{i=1}^m(|a_i - a'_i| + \|w_i - w'_i\|).$$

*Proof.* The proof is based on the straightforward calculation. As for the first inequality, for every $i \in [m]$, it holds that

$$
\left| \bar{a}_i I(\bar{w}_i, \bar{w}_j) \cdot \frac{\mathrm{d}\bar{a}_j}{\mathrm{d}a_j} - \bar{a}_i' I(\bar{w}_i', \bar{w}_j') \cdot \frac{\mathrm{d}\bar{a}_j'}{\mathrm{d}a_j} \right|
$$

$$
\leq \left| \left( \bar{a}_i I(\bar{w}_i, \bar{w}_j) - \bar{a}_i' I(\bar{w}_i, \bar{w}_j) \right) \cdot \frac{\mathrm{d}\bar{a}_j}{\mathrm{d}a_j} \right| + \left| \left( \bar{a}_i' I(\bar{w}_i, \bar{w}_j) - \bar{a}_i' I(\bar{w}_i, \bar{w}_j') \right) \cdot \frac{\mathrm{d}\bar{a}_j}{\mathrm{d}a_j} \right|
$$

$$
+ \left| \left( \bar{a}_i' I(\bar{w}_i, \bar{w}_j') - \bar{a}_i' I(\bar{w}_i', \bar{w}_j') \right) \cdot \frac{\mathrm{d}\bar{a}_j}{\mathrm{d}a_j} \right| + \left| \bar{a}_i' I(\bar{w}_i', \bar{w}_j') \cdot \left( \frac{\mathrm{d}\bar{a}_j}{\mathrm{d}a_j} - \frac{\mathrm{d}\bar{a}_j'}{\mathrm{d}a_j} \right) \right|
$$

$$
\leq |\bar{a}_j - \bar{a}_j'| \frac{\|\bar{w}_i\| \|\bar{w}_j\|}{2d} \cdot 4R + \|\bar{w}_j - \bar{w}_j'\| \frac{dR^2}{2d} 4R + \|\bar{w}_i - \bar{w}_i'\| \frac{dR^2}{2d} 4R + R \frac{\|\bar{w}_i\| \|\bar{w}_j\|}{2d} \cdot |\bar{a}_j - \hat{a}_j|
$$

$$
\leq \frac{5R^3}{2} \left( |a_j - a_j'| + \|w_j - w_j'\| \right) + 2R^3 \|\bar{w}_i - \bar{w}_i'\|,
$$

and

$$
\left| a_i^\circ I(w_i^\circ, \bar{w}_j) \cdot \frac{\mathrm{d}\bar{a}_j}{\mathrm{d}a_j} - a_i^\circ I(w_i^\circ, \bar{w}_j') \cdot \frac{\mathrm{d}\bar{a}_j'}{\mathrm{d}a_j} \right|
$$

$$
\leq \left| \left( a_i^\circ I(w_i^\circ, \bar{w}_j) - a_i^\circ I(w_i^\circ, \bar{w}_j') \right) \cdot \frac{\mathrm{d}\bar{a}_j}{\mathrm{d}a_j} \right| + \left| a_i^\circ I(w_i^\circ, \bar{w}_j') \cdot \left( \frac{\mathrm{d}\bar{a}_j}{\mathrm{d}a_j} - \frac{\mathrm{d}\bar{a}_j'}{\mathrm{d}a_j} \right) \right|
$$

$$
\leq \frac{1}{2d} \|w_j - w_j'\| 4R + \frac{\sqrt{d}R}{2d} |\bar{a}_j - \hat{a}_j| \leq \frac{2\sqrt{d}R}{d} \left( |a_j - a_j'| + \|w_j - w_j'\| \right),
$$

where we use $\|w_j^\circ\| \leq \|W^o\| \leq 1$ for any $j \in [m]$. Then the triangle inequality gives the first assertion. As for the second inequality, for every $i \in [m]$, it holds that

$$
\left\| \bar{a}_i \bar{a}_j J(\bar{w}_i, \bar{w}_j) \odot \frac{\mathrm{d}\bar{w}_j}{\mathrm{d}w_j} - \bar{a}_i' \bar{a}_j' J(\bar{w}_i', \bar{w}_j') \odot \frac{\mathrm{d}\bar{w}_j'}{\mathrm{d}w_j} \right\|
$$

$$
\leq \left\| \left( \bar{a}_i \bar{a}_j J(\bar{w}_i, \bar{w}_j) - \bar{a}_i' \bar{a}_j' J(\bar{w}_i, \bar{w}_j) \right) \odot \frac{\mathrm{d}\bar{w}_j}{\mathrm{d}w_j} \right\| + \left\| \left( \bar{a}_i' \bar{a}_j' J(\bar{w}_i, \bar{w}_j) - \bar{a}_i' \bar{a}_j' J(\bar{w}_i, \bar{w}_j') \right) \odot \frac{\mathrm{d}\bar{w}_j}{\mathrm{d}w_j} \right\|
$$

$$
+ \left\| \left( \bar{a}_i' \bar{a}_j' J(\bar{w}_i, \bar{w}_j') - \bar{a}_i' \bar{a}_j' J(\bar{w}_i', \bar{w}_j') \right) \odot \frac{\mathrm{d}\bar{w}_j}{\mathrm{d}w_j} \right\| + \left\| \bar{a}_i' \bar{a}_j' J(\bar{w}_i', \bar{w}_j') \odot \left( \frac{\mathrm{d}\bar{w}_j}{\mathrm{d}w_j} - \frac{\mathrm{d}\bar{w}_j'}{\mathrm{d}w_j} \right) \right\|
$$

$$
\leq R |a_j - a_j'| \frac{\sqrt{d}R}{2d} \cdot 4\sqrt{d}R + R |a_i - a_i'| \frac{\sqrt{d}R}{2d} \cdot 4\sqrt{d}R + R^2 \frac{\sqrt{d}}{2d} \cdot \|w_j - w_j'\| \cdot 4\sqrt{d}R
$$

$$
+ R^2 \frac{\sqrt{d}}{2d} \cdot \|w_i - w_i'\| \cdot 4\sqrt{d}R + 2R^2 \frac{\sqrt{d}R}{2d} \cdot \|w_j - w_j'\|
$$

$$
\leq 2R^3 \left( |a_j - a_j'| + \|w_j - w_j'\| \right) + 2R^3 \left( |a_i - a_i'| + \|w_i - w_i'\| \right).
$$

and

$$
\left\| \bar{a}_i a_j^\circ J(\bar{w}_i, w_j^\circ) \odot \frac{\mathrm{d}\bar{w}_j}{\mathrm{d}w_j} - \bar{a}_i' a_j^\circ J(\bar{w}_i', w_j^\circ) \odot \frac{\mathrm{d}\bar{w}_j'}{\mathrm{d}w_j} \right\|
$$

$$
\leq \left\| \left( \bar{a}_i a_j^\circ J(\bar{w}_i, w_j^\circ) - \bar{a}_i' a_j^\circ J(\bar{w}_i, w_j^\circ) \right) \odot \frac{\mathrm{d}\bar{w}_j}{\mathrm{d}w_j} \right\| + \left\| \left( \bar{a}_i' a_j^\circ J(\bar{w}_i, w_j^\circ) - \bar{a}_i' a_j^\circ J(\bar{w}_i', w_j^\circ) \right) \odot \frac{\mathrm{d}\bar{w}_j}{\mathrm{d}w_j} \right\|
$$

$$
+ \left\| \bar{a}_i' a_j^\circ J(\bar{w}_i', w_j^\circ) \odot \left( \frac{\mathrm{d}\bar{w}_j}{\mathrm{d}w_j} - \frac{\mathrm{d}\bar{w}_j'}{\mathrm{d}w_j} \right) \right\|
$$

$$
\leq \frac{1}{2d} |a_j - a_j'| 4R + R \frac{1}{2d} \cdot \|w_j - w_j'\| + \frac{R}{2d} \cdot \|w_j - w_j'\| = \frac{2R}{d} \left( |a_j - a_j'| + \|w_j - w_j'\| \right)
$$

Again by using the triangle inequality, we obtain the conclusion. $\qquad \square$

*proof of Lemma C.5.* By using Lemma E.5,

$$\left\|\nabla\mathcal{L}(\theta) - \nabla\mathcal{L}(\theta')\right\|$$

$$\leq \sum_{j=1}^{m} \left(\left|\nabla_{a_j}\mathcal{L}(\theta) - \nabla_{a_j}\mathcal{L}(\theta')\right| + \left\|\nabla_{w_j}\mathcal{L}(\theta) - \nabla_{w_j}\mathcal{L}(\theta')\right\|\right)$$

$$\leq m\left[\frac{5R^3}{2} + \frac{2\sqrt{d}R}{d} + 2R^3 + \frac{2R}{d} + \frac{1}{2}\cdot 2R^3 + 2R^3\right]\sum_{j=1}^{m}\left(\left|a_j - a'_j\right| + \left\|w_j - w'_j\right\|\right)$$

$$\leq m\left[\frac{5R^3}{2} + \frac{2\sqrt{d}R}{d} + 2R^3 + \frac{2R}{d} + \frac{1}{2}\cdot 2R^3 + 2R^3\right]\sqrt{2m}\left\|\theta - \theta'\right\| = L'\left\|\theta - \theta'\right\|$$

holds with $L' = O(m^2 R^3)$. Combining this with the fact that the mapping $\theta \mapsto \lambda\|\theta\|^2$ is $2\lambda$-smooth and the triangle inequality, we obtain that $\mathcal{R}_\lambda(\cdot)$ is $L$-smooth with $L = O(m^2 R^3 + \lambda)$, which gives the conclusion. $\qquad\square$

