# OpenReview forum: "Excess Risk of Two-Layer ReLU Neural Networks in Teacher-Student Settings and its Superiority to Kernel Methods"
_ICLR.cc/2023/Conference — ICLR 2023 poster_

### Official Review · Reviewer_51MF · 2022-10-17

**Confidence:** 3
**Correctness:** 4
**Technical Novelty And Significance:** 3
**Empirical Novelty And Significance:** 3
**Recommendation:** 8

**Clarity, Quality, Novelty And Reproducibility:**

The paper is very clear. All parameters necessary for Fig.1 seem to be specified. Although I'm not extremely familiar with the literature in the field, the results seem of solid interest.

**Strength And Weaknesses:**

Strengths:

The paper is overall well-written, clear and easy to follow. The related works are very satisfyingly discussed.  The question asked (when, and why do neural net outperform linear methods) is definitely an important one, and the setting the authors study is very natural. I have not read the proofs.

Weaknesses:

A plot with the learning curves of the student network, along with those of some kernel methods (RF. NTK, or usual kernels such as RBF), with the relevant upper/lower rates of decay, would be extremely helpful in visualizing (and bolstering) the theoretical results.

Questions:

To my understanding, the bound for the student network seems to hold after a threshold of the order of exp(m**2) (theorem 4.5), which can be extremely large already for widths 50, thereby raising the question of whether the regime described can be observed at all unless the network is very narrow. More discussion in this sense would be helpful (or, in the case I am misunderstanding, a clarification from the authors). Note that this is more a clarification question and in any case the results remain of interest.

For small m (e.g. m=2), does the student net perform clearly better than the linear methods right after the threshold in n predicted in Theorem 4.5? For large dimensions d, I would naively expect from the results of Aubin et al. (The committee machine....., 2019) that there exists a large range of n where this is not the case.

**Summary Of The Paper:**

The authors consider learning from a two-layer network, using a) a student with matching architecture trained with a two-phase Langevin+GD procedure b) linear methods. They provide upper (resp. lower) bounds on excess test error with the number of samples, and show that the student net always display faster rates than linear method. Their discussion also provides some insight about the role of convexity (or lack thereof) to rationalize these findings.

**Summary Of The Review:**

Although I have not reviewed the proof, the paper comes across as clear, well-written, and seems to my understanding to bring solid results, I am therefore giving a good score, but with average-low confidence.

The paper does not display to me a clear weakness, my only comment being that a plot which illustrates the main point of the paper would to my mind improve on the clarity and impact.

---

> ### Author Response · Authors · 2022-11-17
> **Reply to Reviewer 51MF**
>
> Thank you very much for your positive feedback and insightful comments.
>
> 1. A plot with the learning curves of the student network, along with those of some kernel methods (RF. NTK, or usual kernels such as RBF), with the relevant upper/lower rates of decay, would be extremely helpful in visualizing (and bolstering) the theoretical results.
>
> A. Thank you for your insightful suggestion. We do not have all the results now, so we would like to add further experimental results in a camera-ready version.
>
> 2. To my understanding, the bound for the student network seems to hold after a threshold of the order of exp(m**2) (theorem 4.5), which can be extremely large already for widths 50, thereby raising the question of whether the regime described can be observed at all unless the network is very narrow.
>
> A. Thank you for pointing out an important issue. Our setting assumes the “sparse” function class in the sense that $m$ is not too large and at most $d$. To remove the exponential dependency, we need to develop a substantially new convergence analysis for the Langevin dynamics, which we feel is far from trivial. On the other hand, there is a possibility that $m$ could be made small and less problematic when the “degrees of freedom” of the true functions are small even for a large teacher network (please see [1], for example).
>
> [1] Francis Bach: On the Equivalence between Kernel Quadrature Rules and Random Feature Expansions. JMLR, 2017.
>
> 3. For small m (e.g. m=2), does the student net perform clearly better than the linear methods right after the threshold in n predicted in Theorem 4.5?
>
> A. As we have shown in Theorem 4.1 and Theorem 4.5, the student network outperforms linear methods in larger n (in other words, where we require a more precise estimation for the student network). As for your concern about the existing result [1], their negative result occurs when we use the AMP algorithm because of the presence of local minima to the best of our understanding. Indeed, they also derive the small generalization error based on an information-theoretic approach. Our analysis shows that the two-stage optimization escapes bad local minima and eventually achieves good generalization performance. Thus we think our results do not contradict those in [1].
>
> [1] Aubin, Benjamin, et al. "The committee machine: Computational to statistical gaps in learning a two-layers neural network." Advances in Neural Information Processing Systems 31 (2018).

---

### Official Review · Reviewer_xk6b · 2022-10-22

**Confidence:** 4
**Correctness:** 2
**Technical Novelty And Significance:** 3
**Empirical Novelty And Significance:** Not applicable
**Recommendation:** 6

**Clarity, Quality, Novelty And Reproducibility:**

This paper is well written, and has good quality, but the obtained results and claims appear a little specious. Apart from the above drawback, what’s the difference between the current problem setting and the mean field regime in terms of the function space and the obtained gradient based algorithm?


**Strength And Weaknesses:**

**Pros:**

1. propose a two-stage gradient descent algorithm combining the gradient Langevin dynamics and standard gradient descent over the first layer for neural networks training

2. Study the separation between kernel methods and neural networks in terms of excess risk


**Cons:**

1. The authors claimed that their results do not require high over-parameterization conditions. However, according to Theorem 4.5, the convergence rate is $O(m^5/n)$, as well as the minimum singular value $\sigma_{min}$ heavily depending on $m$.
That means, their results are only valid to $m \ll n$ rather than over-parameterization.
Furthermore, I don’t agree with the authors’ current claimed $O(1/n)$ rate for neural networks, because this is a width-dependent result though the authors set $m < d$ in this work.
Instead, kernel methods obtain a width-independent $O(n^{-1/2})$ convergence rate, e.g., (Bach, JMLR 2017, Gal et al. NeurIPS 2022). Hence, in this case, it is not enough to show the separation between kernel methods and neural networks.

[1] Vardi, Gal, Ohad Shamir, and Nathan Srebro. "The Sample Complexity of One-Hidden-Layer Neural Networks." NeurIPS 2022.

2. If we put the above key issue aside, a small $m$ (as well as the experiments) makes sense for the teacher network with few activations.  But for the student network, a few $m$ significantly restrict the model expressivity. I'm wondering the size of function space (in my next question). Besides, this also effects the convergence of phase II in Proposition 4.4 with $O(1/m^3)$. The convergence results require to work in a large $m$ setting, falling into a self-contradiction suitation.

3. The considered function space in Eq. (1) is a norm-based function space, and can be regarded as a subspace of variational space for two-layer neural networks via signed measure (or similar to the path-norm based space), see (Bach, JMLR2017). The separation between kernel methods and two-layer neural networks has been studied in this larger space [2] as kernel methods suffer from the curse of dimensionality $\Omega(n^{-1/d})$.

[2] Parhi, Rahul, and Robert D. Nowak. "Near-minimax optimal estimation with shallow ReLU neural networks." IEEE Transactions on Information Theory (2022).

While this work focuses on a smaller function space and kernel methods are close to the $O(n^{-1/2})$ rate in high dimensional settings. I understand the authors aim to show the separation between kernel methods and neural networks on the convergence rate: $O(n^{-1/2})$ vs. $O(1/n)$. However, strictly speaking, this is not the claimed “curse of dimensionality” in the approximation theory view.

4. Proposition 4.3 appears vacuous due to the exponential order of $m$ and $\beta$. Regarding $\beta$, the authors suggest using “a sufficiently large $\beta$” in phase I to ensure a near-optimal result. However, it makes the first term exponentially large in the first term of Proposition 4.4, resulting in a vacuous result.



**Summary Of The Paper:**

This paper studies the excess risk of two-layer ReLU neural networks training by two-stage gradient based algorithms with spherical data iid spread on the unit sphere. Under such neural networks training, the authors demonstrate that, two layer neural networks are able to achieve the minimax $O(1/n)$ rate in a certain norm-based space, when compared to kernel methods with $O(n^{-1/2})$ rate (slightly faster  than this rate via the convex argument technique).

**Summary Of The Review:**

This work presents the separation between kernel methods and neural networks in terms of excess risk, but the problem setting with a few $m$, sufficiently large $\beta$ in Proposition 4.3, and the function space require more discussion and refinement for publication. I suggest 1) the authors center around a few $m$ for the teacher network, but the student network allows for more neurons;
2) fix the over-parameterization issue in the excess risk;
3) avoid an exponential order of a sufficiently large $\beta$ in Proposition 4.3.

---

> ### Author Response · Authors · 2022-11-17
> **Reply to Reviewer xk6b (1)**
>
> Thank you for your insightful feedback and comments.
> First, we would like to give comments on the weakness that you pointed out.
>
> 1. According to Theorem 4.5, the convergence rate is $ O(m^5/n)$, as well as the minimum singular value  $O(m^5/n)$, heavily depending on $m$. That means, their results are only valid to $m\ll n$  rather than over-parameterization.
>
> A. Thank you for pointing out the important issue. Indeed, our current result can not cover over-parameterized settings seen in recent deep learning literature. However, we expect that in overparameterized settings (in which a student is larger than a teacher), a similar result also holds and would be guaranteed, while we still assume that the teacher width is not greater than the dimensionality. Indeed, the convergence in the first phase (Gradient Langevin dynamics) can be ensured by replacing the term $m$ in Proposition 4.3 with the width of the student. On the other hand, the theoretical difficulty exists in the second phase (the local convergence), and the global convergence is guaranteed if we can ensure the local convergence in over-parameterized settings.
>
> 2. Instead, kernel methods obtain a width-independent $O(n^{−1/2})$ convergence rate,
> e.g., (Bach, JMLR 2017, Gal et al. NeurIPS 2022). Hence, in this case, it is not enough to show the separation between kernel methods and neural networks.
>
> A. We would like to remark that (Bach, JMLR 2017) gives the bound for the generalization gap by using the Rademacher complexity argument. In this research, we focus on the “excess risk”, which is different from the generalization gap.
> Aside from the generalization gap, the local Rademacher complexity argument is required for the excess risk bound (Please see [1], section 13 for example), which utilizes the strong convexity of the loss function for obtaining the faster convergence rate. Indeed, when considering the convergence rate with respect to n, the generalization gap generally converges to zero with $O(1/n^{1/2})$, which is known to be optimal, but our obtained result for the linear estimator (in Theorem 4.1) is faster than that rate.
>
> [1] Martin J. Wainwright. High-Dimensional Statistics: A Non-Asymptotic Viewpoint. Cambridge Series in Statistical and Probabilistic Mathematics. Cambridge University Press, 2019.
>
> 3. Besides, this also effects the convergence of phase II in Proposition 4.4 with O(1/m^3).. The convergence results require to work in a large $m$ setting, falling into a self-contradiction situation.
>
> A. As you mentioned, the small $m$ restricts the expressive power of the student network. On the other hand, we would like to emphasize that the result in Proposition 4.4 is not a pessimistic one hurt by this fact. Proposition 4.4 does not represent the convergence limit in Phase II.  This represents the necessary condition for ensuring the start of the local convergence in Phase II, which eventually archives the convergence rate stated in Theorem 4.5. The noisy gradient descent in Phase I (with hyperparameters stated in Theorem 4.5) is conducted to achieve this necessary condition. We have modified the construction of Section 4.2 to clarify that point; please see Lemma 4.5 in the revised version.
>
> 4. The considered function space in Eq. (1) is a norm-based function space, and can be regarded as a subspace of variational space for two-layer neural networks via signed measure (or similar to the path-norm based space), see (Bach, JMLR2017). The separation between kernel methods and two-layer neural networks has been studied in this larger space [2] as kernel methods suffer from the curse of dimensionality $\Omega(n^{-1/d})$.

---

> > ### Author Response · Authors · 2022-11-17
> > **Reply to Reviwer xk6b (2)**
> >
> > (This is a continuation of "Reply to Reviwer xk6b (1)". We are sorry for the long reply.)
> >
> > A. Thank you for pointing out our related work. We would like to argue that there is a difference between our result and [2]. First, our objective is to derive the superiority of ReLU neural networks to linear estimators with an optimization guarantee (in other words, the superiority of NNs trained by a gradient method). [2] only shows the existence of the parameters of the neural network that achieve the min-max optimal learning rate and does not consider the optimization process. Moreover, we would like to emphasize that the superiority of neural networks does not always hold when we consider the subclass of the variational space. Since we consider the much narrower class as the true model than [1] and [2], whether the curse of dimensionality will occur in linear estimators or not is not trivial. Indeed, the derived learning rates (in Theorem 4.1 and Theorem 4.5) are different from those derived in [2]. While the obtained lower bound for linear estimators ($O(n^{-(d+2)/(2d+2)})$) is faster than what is obtained in [2] ($O^({-3/(d+3)})$), the obtained upper bound for the student network ($O(m^5/n)$) is improved by the lower bound obtained in [2] by considering the narrower function space for the true model.
> > Moreover, we would like to mention that while (Bach, JMLR2017) describes $n^{−1/d}$ as the lower bound of the approximation error of neural networks,  $n$ means the network width (i.e., $m$ in this paper).
> >
> > 5. Proposition 4.3 appears vacuous due to the exponential order of $m$ and $\beta$. Regarding $\beta$, the authors suggest using “a sufficiently large $\beta$” in phase I to ensure a near-optimal result. However, it makes the first term exponentially large in the first term of Proposition 4.4, resulting in a vacuous result.
> >
> > A. Thank you for pointing out the ambiguous point in our writing. Proposition 4.4 is the result that holds independently of Proposition 4.3 whenever the objective value ($R_\lambda$) is sufficiently small. Then, the necessary condition for ensuring the local convergence in phase II is derived by taking $\beta$ and other hyperparameters to achieve the condition $R_\lambda-R^*\le \epsilon$ based on Proposition 4.3. Please see Theorem 4.5, in which the overall setting of each hyperparameter to obtain the near-optimal rate is summarized.
> >
> > Moreover, we would like to answer your question in “Clarity, Quality, Novelty And Reproducibility”.
> >
> > 6. Apart from the above drawback, what’s the difference between the current problem setting and the mean field regime in terms of the function space and the obtained gradient based algorithm?
> >
> > A. As you mentioned in the above comment, the function space in the current problem is included in that of the mean field regime. Our work provides the “polynomial time” optimization algorithm by utilizing the structure of the true model (teacher network), while there are various results ensuring the global convergence of the neural networks in the mean field regime.
> >
> > Finally, we would like to comment on your suggestion in “Summary Of The Review”.
> >
> > 7.  I suggest 1) the authors center around a few $m$ for the teacher network, but the student network allows for more neurons; 2) fix the over-parameterization issue in the excess risk; 3) avoid an exponential order of sufficiently large $\beta$ in Proposition 4.3.
> >
> > A. We are grateful for your suggestion. We also feel that your points are all important future work to be resolved. As for 1) and 2),  we expect that in overparameterized settings, a similar result also holds and would be guaranteed, as we stated in your comment 1.. As for 3), the exponential dependence on $\beta$ originates from the evaluation of the log-Sobolev constant $\alpha$ defined in Lemma C.4. We utilize Lemma E.4 for its derivation, but there may be room for improvement indeed. However, it is required to investigate the landscape of the objective function for improvement, and we feel it is challenging.

---

> > ### Comment · Reviewer_xk6b · 2022-11-17
> > **increase my score to 5 but the neural network width $m$ setting falls into a dilemma**
> >
> > Thanks for the authors' feedback.
> >
> > 1. **over-parameterization:** acutally it's ok for me that this work focuses on a under-parameterzied regime. However, the authors mentioned that, the over-parameterized regime works in which a student is larger than a teacher. It's true that there exists some different definitions on over-parameterization, e.g., the number of parameters is larger than the number of training data or the input dimension. But it's unclear to me the case that a student is larger than a teacher belongs to the over-parameterized regime as I don't find similar claim in previous literature.
> >
> > 2. **convergence rate in kernel methods and variational norm based space:** I agree with the authors' convex argument that achieves slightly better convergence rate than $O(n^{-1/2})$. Besides, [2] gives the convergence rate at $O(n^{-\frac{d+3}{2d+3}})$, which is faster than the obtained rate in this work. It's unclear to me why the authors mentioned the convergence rate of [2] at O(n^{-d/(d+3)}). maybe I'm missing something?
> >
> > When reading the authors' response, I think the contribution reuqires a better organization when compared to [1,2] in the variational norm based space.
> >
> > **Though a smaller space is considered, this work provides a computational efficient optimization algorithm beyond RKHS. Besides, a sepration between nerual networks and kernel methods is conducted from the perspective of convergence rate in the excess risk.**
> >
> > I suggest the authors to center around this point in the introduction as this is the main contribution and message to the readers.
> > If the two-stage gradient descent algorithm makes sense, I will vote for accept but I'm unsatisfactory with the derived bound in Prop. 4.3. See the next point.
> >
> > 3. **exponential order of $\beta$ and $m$**: This is the main reason that I vote for reject of this paper because the main contribution is to develop a computational efficient optimization algorithm beyond RKHS. In this case, I don't believe the authors' claim $R_\lambda - R^* \leq \epsilon$ due to the exponential dependency of $m$ and $\beta$. This requires $m$ is small but restricts the express the representation ability of neural networks. When reading Theorem 4.5 and Algorithm 1, the inverse temperature $\beta$ makes sense.
> >
> > Based on the above discussion, I increase my score to 5 but still care about the exponential order of $m$ and the less expressive ability under a small $m$ (fall into a dilemma). I suggest the authors to discuss these drawback, compare with [1,2], re-organize the contribution.

---

> > > ### Author Response · Authors · 2022-11-18
> > > **Thank you for your response**
> > >
> > > Thank you very much for your prompt reply and increasing your score.
> > > Please find our reply to your additional comments.
> > >
> > > 1. It's unclear to me the case that a student is larger than a teacher belongs to the over-parameterized regime as I don't find similar claim in previous literature.
> > >
> > > A. We intend to state “over-parameterized” as “larger than the width sufficient to represent the true model.” While that definition is different from those seen in the NTK (or other over-parameterized models) literature, the same can be seen in the teacher-student setting, such as [1].
> > >
> > > [1] Itay M Safran, Gilad Yehudai, and Ohad Shamir. The effects of mild over-parameterization on the optimization landscape of shallow relu neural networks. In Conference on Learning Theory, pp. 3889–3934. PMLR, 2021.
> > >
> > > 2. Besides, [2] gives the convergence rate at $O(n^{−d+3/2d+3})$,  which is faster than the obtained rate in this work. It's unclear to me why the authors mentioned the convergence rate of [2] at O(n^{-d/(d+3)}). maybe I'm missing something?
> > >
> > > A. We would like to note that the convergence rate at $O(n^{−d+3/2d+3})$ is not faster than the rate $O(1/n)$ (when $m$ is not too large), which is obtained in this work.
> > > Our objective to mention $O(n^{-3/(d+3)}$) (we guess you are referring to this rate , not $O(n^{-d/(d+3)}))$ is the comparison of the lower bound of the linear estimators (Theorem 4.1) with that obtained in [2].
> > >
> > > 3. exponential order of $\beta$ and $m$
> > >
> > > A.
> > >
> > > (About $\beta$) Indeed, $\beta$ affects exponentially the convergence rate in Phase I. To the best of our knowledge, this exponential order is unavoidable by standard analysis of Langevin dynamics, and any existing literature does not clear this point. On the other hand, we can take not too large $\beta$ (constant with respect to $n$) by considering the two-stage algorithm. This is the reason why we employ Algorithm 1 (the two-stage algorithm).
> > >
> > > (About $m$) This exponential dependency is merely due to the evaluation of the log-Sobolev constant (Lemma C.4), in which $\alpha$ depends exponentially on $m$ since the sup-norm of the student network depends on $m$. Therefore, we can indeed remove this dependency by considering the following settings: (1) utilizing the mean field network (multiplying 1/m to the output of the student), (2) $w_j$s are directed to different directions to some extent, and hence the sup-norm can be bounded. In a practical setting, it would be natural to assume the teacher network is bounded. In that case, $\mathrm{exp}(m)$ does not appear, which would partially explain the practical success of neural networks. We have added these points as the remark after Lemma C.4. Otherwise, it is quite hard to remove the exponential dependency, and thus we considered the sparse model as the teacher, i.e., $m$ is not too large.

---

> > > > ### Comment · Reviewer_xk6b · 2022-11-18
> > > > **comparison with [2] and exponential order**
> > > >
> > > > Thanks for the authors' clarification.
> > > >
> > > > In fact, comparison with [2] in the author response is relatively unfair. This work studies (upper and lower) bound of excess risk and obtain faster rates than [2] but works in a smaller function space. I think this requires a care comparison and discussion in the paper.
> > > >
> > > > I understand the difficulty of avoiding exponential order of $\beta$, but its dependency on $\Omega(d)$ (see in Theorem 4.6) doesn't convince me as the authors claim previous optimization algorithms in the mean field regime also suffer from the curse of dimensionality. Acutally I'm not sure that the men field regime can avoid the exponential order of $m$. If this is true, the difference between the current function space/algorithms and that in the mean field regime requires a detailed discussion (my previous question mention this).

---

> > > > > ### Author Response · Authors · 2022-11-18
> > > > > **Thank you for clarifying what you mentioned**
> > > > >
> > > > > Thank you very much for your insightful comments that address the details of our research.
> > > > >
> > > > > *Comparison with [2]:* Our explanations about the comparison with [2] would have been a bit rough. Here, we would like to clarify the details again. First, we do not intend that our result is "better" than [2], but they are not directly comparable with ours because we consider a narrower function class than [2]. We note that there are two types of minimax optimal rates, that is, the one is the optimal rate over all estimators and the other is that over the class of linear estimators. In [2], they consider a bounded variation norm class in which the minimax rate over all estimators is $O(n^{-(d+3)/(2d+3)})$ and that over the linear estimators is $O(n^{-d/(d+3)})$. On the other hand, in our setting, we obtained an upper bound $O(m^5/n)$ for the excess risk of our method and the minimax rate over the linear estimators as $O(n^{-(d+2)/(d+2)})$. So, roughly $O(n^{-(d+3)/(2d+3)})$ in [2] can be compared with our rate $O(m^5/n)$, and $O(n^{-d/(d+3)})$ in [2] can be compared with our rate $O(n^{-(d+3)/(2d+3)})$. But, again we should note that their target function class is very different from ours.
> > > > >
> > > > > *Connection to mean field regime:* Thanks so much for raising an interesting discussion about connection to the mean field regime. It is easy to see the exponential dependency on $m$ can be avoided by employing the mean field scaling, that is, using $O(1/m)$ weight on each neuron. Indeed, the exponential dependence on $m$ stems from the Holly-Stroock bounded perturbation argument (Lemma E.4) where the sup-norm of the student network affects the rate in the exponential order through the log-Sobolev constant. In our setting, the sup-norm of the student network depends on $m$ which induces the exponential dependency on $m$. Therefore, if the sup-norm of the student network is bounded, we can avoid the exponential dependency; which can be accomplished by dividing the output of the student network by $m$ (which we have meant by the "mean field regime") or clipping the output by applying something like $\tanh$ function so that it becomes bounded. Here, we would like to emphasize that we do not consider the setting where $m\to \infty$ (which is a possible setting in the mean field regime) in this work. We only focus on the setting where $m$ is fixed (as a finite number). Sorry for misleading writing with the term "mean field regime."
> > > > >
> > > > > *(dependency on $d$)* On the other hand, it is also interesting to compare with the mean field Langevin dynamics in terms of dependency on $d$. The exponential dependency of $\beta$ on $d$ is unavoidable as long as we are using the standard analysis of the gradient Langevin dynamics because there exists $O(d\log(\beta)/\beta)$ term in the optimization error (see for example Raginsky et al. (2017)). There are no results overcoming this problem to the best of our knowledge.
> > > > > As you pointed out, it would be worth discussing whether the mean field Langevin avoids dependency on $d$. Chizat (2022) and Nitanda et al. (2022) showed that the mean field Langevin dynamics optimizes an objective that is the sum of the loss and the entropy term: $L(\mu) + 1/\beta \mathrm{Ent}(\mu)$. This bound looks dimension independent, however the entropy term implicitly depends on the dimension. For example, if $\mu$ (the distribution of parameters) is Gaussian, the entropy term is $\Omega(d)$. Hence, to make this term below the threshold, the inverse temperature $\beta$ should be $\Omega(d)$ which exponentially affects the rate of convergence through the log-Sobolev constant. Therefore, the mean field Langevin dynamics also suffers from exponential dependency on $d$ as long as we use the existing result. Moreover, we need to set at least $\beta = O(n)$ so that the excess risk becomes $O(1/n)$ by the vanilla mean field Langevin. This induces exponential dependency on $n$. Our two phase algorithm is beneficial because we can avoid this dependency.
> > > > >
> > > > > Maxim Raginsky, Alexander Rakhlin, Matus Telgarsky: Non-convex learning via Stochastic Gradient Langevin Dynamics: a nonasymptotic analysis. Proceedings of the 2017 Conference on Learning Theory, PMLR 65:1674-1703, 2017.
> > > > >
> > > > > Lénaïc Chizat: Mean-Field Langevin Dynamics : Exponential Convergence and Annealing. TMLR, 2022.
> > > > >
> > > > > Atsushi Nitanda, Denny Wu, Taiji Suzuki: Convex Analysis of the Mean Field Langevin Dynamics. Proceedings of The 25th International Conference on Artificial Intelligence and Statistics, PMLR 151:9741-9757, 2022.

---

> > > > > > ### Comment · Reviewer_xk6b · 2022-12-12
> > > > > > **increase my score to 6**
> > > > > >
> > > > > > I think avoiding the curse of dimensionality (CoD) in optimization under the variational space appears impossible. The CoD occurs in either the input dimension $d$ or the temperature parameter $\beta$, so I increase my score to 6 as the (main) target in the optimization part is to design a gradient-based algorithm in this space, which can be complementary to Pilanci's work in optimization.

---

### Official Review · Reviewer_Xjf3 · 2022-10-24

**Confidence:** 3
**Correctness:** 3
**Technical Novelty And Significance:** 3
**Empirical Novelty And Significance:** Not applicable
**Recommendation:** 5

**Clarity, Quality, Novelty And Reproducibility:**

Other than Section 4.2, the paper is clearly written. However, I need to say the novelty is limited.

**Strength And Weaknesses:**

### Strength:
* The conclusion on the superiority of the neural networks is interesting.

### Weakness:
* The presentation of the paper, especially Section 4.2 should be improved. The two-phase algorithms are not presented in a unified way, which leads to some ambiguities for me.
* The results are not a substantial improvement over [1]. The only new result for this paper is showing that when the teacher and the student have the same width, we can have a computationally efficient algorithm to perform the estimation, using the idea from [2].

[1] Suzuki, Taiji, and Shunta Akiyama. "Benefit of deep learning with non-convex noisy gradient descent: Provable excess risk bound and superiority to kernel methods." In International Conference on Learning Representations. 2021.

[2] Zhang, Xiao, Yaodong Yu, Lingxiao Wang, and Quanquan Gu. "Learning one-hidden-layer relu networks via gradient descent." In The 22nd international conference on artificial intelligence and statistics, pp. 1524-1534. PMLR, 2019.

**Summary Of The Paper:**

This paper consider the optimization and the statistical guarantee of two-layer ReLU neural networks under the teacher-student settings, demonstrating that we can obtain a fast rate on the excess risk with a computational-efficient two phase optimization algorithm, that can be shown to beat any linear estimators on the statistical efficiency.

**Summary Of The Review:**

The main concern for me is about the novelty. If I understand correctly, the only new content for this paper is the computational efficient algorithm. However, this part is presented poorly. I do have a specific question on this two-phase algorithm. Following the constants selected in Theorem 4.5, we still need $k^{(1)} = \exp(\mathrm{poly}(m))$, which can still be pathological when $m$ is not so small. Is this really satisfactory? Furthermore, 	I’m wondering if Proposition 4.3 is a satisfactory result, as when $\beta$ goes to infinity, the risk does not go to 0.

---

> ### Author Response · Authors · 2022-11-17
> **Reply to Reviewer Xjf3**
>
> Thank you for your insightful feedback and comments.
> We would like to answer your concern about our result.
>
> 1. The presentation of the paper, especially Section 4.2 should be improved. The two-phase algorithms are not presented in a unified way, which leads to some ambiguities for me.
>
>
> A. We are grateful for your pointing out an issue in our writing. We have rewritten Section 4.2 based on your comment. In Particular, we have added the convergence result of Phase II before Theorem 4.5 and separated Theorem 4.5 to another paragraph as “Unified risk bound”. Moreover, we have assigned numbers to each paragraph and the corresponding numbers to the explanation at the beginning of Section 4.2.
>
>
> 2. The results are not a substantial improvement over [1]. The only new result for this paper is showing that when the teacher and the student have the same width, we can have a computationally efficient algorithm to perform the estimation, using the idea from [2].
>
>
> A. First, we would like to emphasize that [1] does not cover the ReLU activation considered in this work. Moreover, [1] assumes the teacher networks with infinite width and decaying importance, as we stated in Remark 4.6. Thus, the neural network model considered in our work is substantially different from [1]. As for the technical contribution, we extend the convergence result of gradient Langevin dynamics to the non-smooth objective function. While we treat the non-smooth objective function \hat{R}_\lambda (which cannot be treated in existing results), we can ensure the convergence of phase I as Proposition 4.3.
>
>
> 3. we still need $k^{(1)}=\mathrm{exp}(\mathrm{poly}(m))$, which can still be pathological when m is not so small. Is this really satisfactory?
>
>
> A. Thank you for pointing out an important issue. Our setting assumes the “sparse” function class in the sense that $m$ is not too large and at most $d$. To remove the exponential dependency, we need to develop a substantially new convergence analysis for the Langevin dynamics, which we feel is far from trivial. On the other hand, there is a possibility that $m$ could be made small and less problematic when the “degrees of freedom” of the true functions are small even for a large teacher network (please see [1], for example).
>
> [1] Francis Bach: On the Equivalence between Kernel Quadrature Rules and Random Feature Expansions. JMLR, 2017.
>
>
> 4. I’m wondering if Proposition 4.3 is a satisfactory result, as when $\beta$ goes to infinity, the risk does not go to 0.
>
>
> A. As we stated after Proposition 4.3, we do not have to let $\beta \to \infty$ because it is sufficient to make the excess risk below a threshold (which is strictly positive) to enter the phase II. Actually, the left-hand side in Proposition 4.3 won’t go to 0 since we evaluate the generalization risk of the student network trained by a finite sample. By using a large sample size (as stated in Theorem 4.5), the influence of the sample noise decreases and we can get a sufficiently small risk for ensuring the local convergence in phase II.

---

### Official Review · Reviewer_vbHH · 2022-10-25

**Confidence:** 3
**Correctness:** 4
**Technical Novelty And Significance:** 3
**Empirical Novelty And Significance:** 3
**Recommendation:** 8

**Clarity, Quality, Novelty And Reproducibility:**

This paper is clear and well-written. The idea is quite interesting, although the application may be limited.

**Strength And Weaknesses:**

Strength:
1. The proposed algorithm can recover the teacher network's parameters, which is a hard problem.
2. This paper is well-written and organized.
3. This paper has proof and experiments to support their results.

Weakness:
1. This paper requires the student network has the same width as the teacher network. It is unknown whether such a result can be generalized to the case where we don't know m in advance.
2. To connect the noisy SGD and vanilla SGD, the algorithm requires reparameterizing the neural network. This kind of trick seems to be impractical in real applications, especially for DNNs.

**Summary Of The Paper:**

This paper studies the excess risk of two-layer ReLU neural networks in a teacher-student regression model. In particular, they showed that the student network could learn the teacher network under certain conditions in two phases: first by noisy gradient descent
and then by vanilla gradient descent.

**Summary Of The Review:**

In general, this paper is well written. My only concern is whether the exact recovery of the parameter is necessary for NN training, if we only care about the excess risk. Instead of reparemetrizing, can we directly decrease the noise level in Noise GD to zero?

---

> ### Author Response · Authors · 2022-11-17
> **Reply to Reviewer vbHH**
>
> Thank you very much for your positive feedback and insightful comments.
> We would like to answer your concern about our result.
>
> 1. This paper requires the student network has the same width as the teacher network. It is unknown whether such a result can be generalized to the case where we don't know m in advance.
>
> A. Thank you for pointing out an important point. We expect that if a student is wider than a teacher (where the student architecture ’covers’ the teacher), the same faster rate can be obtained. We can ensure the convergence in the first phase (Gradient Langevin dynamics) by replacing a part of $m$ in Proposition 4.4 with the width of the student. However, theoretical difficulty exists in the second phase (the local convergence). Global convergence is guaranteed if we can ensure local convergence. We think that the assumption of wider students is consistent with recent ‘’overparameterized’ settings. On the other hand, if the student is narrower than the teacher, the student does not cover the teacher, and the faster rate may not be derived.
>
> 2. To connect the noisy SGD and vanilla SGD, the algorithm requires reparameterizing the neural network. This kind of trick seems to be impractical in real applications, especially for DNNs.
>
> A. Indeed, the reparameterization technique is introduced to make the analysis simple rather than pursuing an absolutely realistic algorithm. We expect that the reparameterization could be removed with a slight modification to the analysis. We remain it a future work, but we feel it is an important problem to be resolved.
>
> 3. My only concern is whether the exact recovery of the parameter is necessary for NN training, if we only care about the excess risk.
>
> A. There are some reasons why we considered the exact recovery. First, under our setting where $W^o$ is column full rank, and the student has the same width as the teacher, small excess risk automatically leads to exact recovery. We agree that if we consider an overparameterized setting, the exact recovery would not be necessary. However, it is a challenging issue, and we would like to defer it to future work. Second, we want to make full use of the structure of the teacher network to yield the polynomial time computational complexity. The exact recovery is a natural device to show the local convergence since we have the local strong convexity.
>
> 4. Instead of reparemetrizing, can we directly decrease the noise level in Noise GD to zero?
>
> A. That is an important future work. As you mentioned, we may be able to consider the annealing procedure instead of two-stage optimization, i.e., reducing the scale of noise (in other words, enlarging $\beta$) through the training procedure.

---

### Official Review · Reviewer_VLSM · 2022-10-25

**Confidence:** 4
**Correctness:** 4
**Technical Novelty And Significance:** 3
**Empirical Novelty And Significance:** Not applicable
**Recommendation:** 8

**Clarity, Quality, Novelty And Reproducibility:**

This paper is written clearly and with high quality. The novelty of its results lies in that it improved the previous analysis to settings where assumptions are more natural.

**Strength And Weaknesses:**

Strengths:
1. This paper is clearly and rigorously written. All the notations are well explained and most of the technical difficulties are introduced naturally. It is very delightful to read as the paper managed to convey these highly technical statements with such clarity and fine elaboration.
2. The story in this paper is complete, in that this paper almost resolves most of the questions in its setting. They have the convergence guarantee with well-explained optimization algorithms and analysis, and also a generalization guarantee as excess risk upper bound. For the lower bound, they proved the LB for all linear classes (kernels) that their generalization over the teacher class is poor compared to student neural nets.
3. The problem this paper deals with is a long-pursued one in the deep learning theory community. The results in this paper go beyond the previous works and are proven for a setting with almost minimal assumptions over the target function class (for example, it goes beyond the orthogonality condition on which many previous results are based, and it goes beyond the local convergence result of Zhou et al. 2021). This is highly non-trivial progress in this direction. The results of this paper also provide some solid technical contributions.

Weaknesses:
1. The two-stage optimization process, which uses two different optimization algorithms, seems to be the remaining problem to solve. It would be better if the analysis can be applied to settings without this technical workaround.


**Summary Of The Paper:**

This paper studies the excess risk of learning neural networks in the teacher-student setting, as well as the theoretical superiority of neural networks over other linear (or kernel) methods. It provides an excess risk bound of $\frac{\log n}{n}$ convergence rate (with polynomial dependency on other parameters) for the neural networks, and also a minimax lower bound of $n^{-\frac{d+2}{2d+2}-o(1)}$ for all kernel methods to learn the same target function class.

**Summary Of The Review:**

This paper presents a solid contribution to the area of deep learning theory, although it has not resolved all the problems under its setting, it still manages to improve the previous analysis and give an almost complete story of learning in the student-teacher setting beyond kernel methods.

---

> ### Author Response · Authors · 2022-11-17
> **Reply to Reviewer VLSM**
>
> Thank you very much for your positive feedback and insightful comments.
> As for your comment on the weakness in our result:
>
> 1. The two-stage optimization process, which uses two different optimization algorithms, seems to be the remaining problem to solve. It would be better if the analysis can be applied to settings without this technical workaround.
>
> A. Thank you for pointing out the remaining problem. As you mentioned, the two-stage optimization process is motivated by a technical reason. We expect that this process will be extended to a more practical process. First, the noisy gradient descent can be seen in practical techniques such as SGD, and phase I could be replaced by such practical methods. On the other hand, the noisy structure observed in SGD is different from Gaussian (which we utilize in this work), and it is a crucial problem to consider what will occur in SGD. Moreover, we may be able to consider the annealing procedure instead of two-stage optimization, i.e., reducing the scale of noise (in other words, enlarging $\beta$) through the training procedure. It is also an important future work.

---

### Decision · Program_Chairs · 2023-01-20

**Decision:**

Accept: poster

**Justification For Why Not Higher Score:**

While this is a very nice paper I did not see any particular reason for a highlight. It is a nice theory paper but the obtained rates are somewhat disappointing for practical dimensional and width. So a poster is the best here I think.



**Justification For Why Not Lower Score:**

Valuable paper that should not be rejected.

**Metareview: Summary, Strengths And Weaknesses:**

Reviewers overall agree this paper should be accepted. I think the reviews summarize very well the strengths and weaknesses of the paper as well as points that the authors should include in the revised version. This is a nice addition to the conference.

**Note From Pc:**

if the above contains the word "oral" or "spotlight" please see: "oral" presentation means -> notable-top-5% and "spotlight" means -> notable-top-25%. As stated in our emails, we are disassociating presentation type from AC recommendations